# Fair Regression via Plug-In Estimator and Recalibration

**Evgenii Chzhen***
LMO, Université Paris-Saclay
CNRS, Inria

**Christophe Denis**
LAMA, Université Gustave Eiffel
MIA-Paris, AgroParisTech
INRAE, Université Paris-Saclay

**Mohamed Hebiri**
LAMA, Université Gustave Eiffel
CREST, ENSAE, IP Paris

**Luca Oneto**
DIBRIS, University of Genoa

**Massimiliano Pontil**
Istituto Italiano di Tecnologia
University College London

## Abstract

We study the problem of learning an optimal regression function subject to a fairness constraint. It requires that, conditionally on the sensitive feature, the distribution of the function output remains the same. This constraint naturally extends the notion of demographic parity, often used in classification, to the regression setting. We tackle this problem by leveraging on a proxy-discretized version, for which we derive an explicit expression of the optimal fair predictor. This result naturally suggests a two stage approach, in which we first estimate the (unconstrained) regression function from a set of labeled data and then we recalibrate it with another set of unlabeled data. The recalibration step can be efficiently performed via a smooth optimization. We derive rates of convergence of the proposed estimator to the optimal fair predictor both in terms of the risk and fairness constraint. Finally, we present numerical experiments illustrating that the proposed method is often superior or competitive with state-of-the-art methods.

## 1 Introduction

During the recent years algorithmic fairness has emerged as a fundamental area of machine learning, due to the potential risk that standard learning algorithms, when trained on sensitive datasets, may inherit or amplify bias present in the data. This has raised the challenge to design novel algorithms that, while still optimizing prediction performance, mitigate or remove unfairness of the learned predictor, see the papers and books [4, 13, 15, 21, 24, 29, 31, 33, 34, 40, 41, 57–60] and references therein. Until very recently, most work has focused on classification problems, with regression receiving far less attention. However regression problems are equally important for algorithmic fairness. For example, both the problems of predicting students' performance without discriminating based on the gender, or predicting the crime risk of a community without discriminating based on the race, can be cast as regression.

In this paper we study the problem of designing computationally efficient and statistically principled learning methods for fair regression. We define the optimal fair regression function as the one that

minimizes the population squared error subject to a fairness constraint that asks that the function output is independent from the sensitive feature. This notion of fairness is referred to as demographic parity and is widely used in the literature, both in the context of classification and regression [2, 14, 28, 32, 47].

The above definition of optimal fair regression function is not well suited to design an efficient algorithm. Therefore, we first consider a proxy-discretized version of the fair regression problem, for which we derive an explicit expression of the optimal fair predictor. Importantly, we show that this discretization scheme does not alter the quality of the optimal rule: the optimal fair predictors for both problems (the discretized and the original one) have close risks, controlled by the discretization parameter. Our expression for the discretized optimal predictor naturally suggests a plug-in two stage approach, in which we first estimate the (unconstrained) regression function from a set of labeled data and then we recalibrate it with another set of unlabeled data. The latter step can be efficiently performed via a smooth optimization.

A key feature of our approach is that it can be employed alongside any off-the-shelf regression learning method and, provided this one is consistent, our recalibration step transforms in a simple way the original (unconstrained) regression estimator into a prediction function which consistently estimates the optimal fair regression function. This strategy is particularly appealing in those applications where the cost of re-training an existing learning algorithm is high. Furthermore, we derive rates of convergence of the proposed estimator to the optimal fair predictor both in terms of the risk and the fairness constraint violation.

Finally, we present numerical experiments with the proposed method on five real datasets, indicating that our method is often superior or competitive with state-of-the-art methods. In particular, when using random forest as the base regression estimator, our approach results in substantial decrease in fairness violation, at the costs of only a moderate increase in the prediction error rate.

**Previous work.** One of the first work on fair regression is [12], where the authors study the problem of linear regression imposing constraints on the mean outcome or residuals of the models (fairness in expectation). More recently, several authors [2, 6, 26, 36, 38, 43, 47–49] focus on the fair regression problem all employing various fairness definitions. Similarly to [12], the works [6, 36, 48] deal with the linear regression setup by refining the definition of fairness. Raff et al. [49] and Fitzsimons et al. [26] examine the incorporation of fairness in expectation constraints in tree based regression methods. Pérez-Suay et al. [48] incorporate a penalty on the dependence between the predictor and the sensitive attribute into the kernel ridge regression formulation. Unlike these contributions, we do not assume neither linear nor linear in a kernel space relationships between the input and the output.

More related to our work are the papers by Oneto et al. [47] and Agarwal et al. [2]. The former introduces a framework for fair Empirical Risk Minimization (ERM) in the context of regression, providing general bounds in the case of fair regression in RKHS, using a relaxed notion of linearized fairness. The latter paper elegantly transforms the problem of bounded fair regression to a classification problem and then employs the reduction approach of [1]. They derive ERM-type generalization guarantees which are applicable to any class of predictors with bounded pseudo-dimension. Two notions of fairness are used, closest to ours being the Kolmogorov-Smirnov (KS) distance. In contrast to the above papers, we measure unfairness by the Total Variation (TV) distance, which is a stronger notion than the KS distance. Furthermore, our guarantees do not require the optimal predictor to be in a Glivenko–Cantelli or a bounded pseudo-dimension class. Yet, the price for such a guarantee is an extra mild assumption on the distribution of the observations.

Our theoretical contribution is partly inspired by recent work of Chzhen et al. [18], where the authors study binary classification using the Equal Opportunity constraint [see 29]. While they also provide a two stage plug-in approach, the setting considered here induces a non-trivial adaptation of their method of proof, involving a discretization step to deal with the uncountable nature of the constraint. Moreover, contrary to them, we derive finite sample bounds.

## 2 Fair regression

In this section, we introduce the fair regression problem and describe a discretized version of it, for which we derive an explicit form of the optimal regression function.

## 2.1 Learning setting

We let $(X, S, Y) \in \mathbb{R}^d \times \mathcal{S} \times \mathbb{R}$ be random tuple distributed according to a Borel probability measure $\mathbb{P}$ on $\mathbb{R}^d \times \mathcal{S} \times \mathbb{R}$. Here $X \in \mathbb{R}^d$ is a feature vector, $S \in \mathcal{S} := \{-1, 1\}$ is a binary sensitive feature (*i.e.,* protected attribute), and $Y \in \mathbb{R}$ is a real valued signal to be predicted. For all $s \in \mathcal{S}$ we denote by $\mathbb{P}_{X|S=s}$ the conditional distribution of $X|S = s$, by $p_s = \mathbb{P}(S = s)$ the marginal distribution of $S$, and by $\eta(X, s) = \mathbb{E}[Y|X, S = s]$ the conditional expectation of $Y$. Throughout the paper, we denote by $\mathcal{F}$ the set of *all* Borel measurable functions $f : \mathbb{R}^d \times \mathcal{S} \to \mathbb{R}$. In this work we study predictors which include $s \in \mathcal{S}$ in their functional form.

We consider the standard mean squared risk of a predictor $f$ defined as $\mathcal{R}(f) := \mathbb{E}(Y - f(X, S))^2$. We consider a natural extension of the Demographic Parity [11] as the notion of fairness[2], which was previously used in the context of regression by [2, 14, 32].

**Definition 2.1** (Fair predictor). *We say that a predictor $f \in \mathcal{F}$ is* fair *with respect to the distribution $\mathbb{P}$ on $\mathbb{R}^d \times \mathcal{S} \times \mathbb{R}$ if for all Borel sets $\mathcal{C} \subset \mathbb{R}$ it holds that*

$$\mathbb{P}\left(f(X, S) \in \mathcal{C} \,|\, S = -1\right) = \mathbb{P}\left(f(X, S) \in \mathcal{C} \,|\, S = 1\right) \ .$$

In other words, a function $f$ is fair if the total variation distance between the two conditional distributions of the function output associated to the two values of the sensitive feature is zero. For any Borel set $\mathcal{C} \subset \mathbb{R}$ and any predictor $f$, we also introduce the shorthand notation

$$\mathcal{U}(f, \mathcal{C}) := \left|\mathbb{P}\left(f(X, S) \in \mathcal{C} \,|\, S = -1\right) - \mathbb{P}\left(f(X, S) \in \mathcal{C} \,|\, S = 1\right)\right| \ . \tag{1}$$

Finally, we define the fair optimal predictor as a minimizer of the risk under the fairness constraint, that is,

$$f^* \in \arg\min_{f \in \mathcal{F}} \left\{ \mathcal{R}(f) \,:\, \sup_{\mathcal{C} \subset \mathbb{R}} \mathcal{U}(f, \mathcal{C}) = 0 \right\} \ . \tag{$\mathcal{P}$}$$

Notice that the feasible set of the problem $(\mathcal{P})$ is non-empty for any distribution $\mathbb{P}$ as it contains all constant predictors.

**Remark 2.2.** *In this work the sensitive attribute $s \in \mathcal{S}$ enters explicitly in the functional form of the predictor. However, in some applications (e.g. in the law domain) this may not be permitted. In Supplementary Material we show how to modify our methodology to address the case when the predictors take the form of $f : \mathbb{R}^d \to \mathbb{R}$.*

Let us also emphasize that, unlike previous theoretical investigations of fair regression [2, 47], we do not restrict $\mathcal{F}$. Throughout this work we pose the following boundedness assumption on the signal $Y \in \mathbb{R}$, which is also made in the above papers.

**Assumption 2.3** (Bounded signal). *There exists $M > 0$ such that $|Y| \leq M$ almost surely.*

The constant $M$ or its upper bound is assumed known a-priori. This knowledge may naturally arise from the specific application at hand, *e.g.,* GPA of a student.

## 2.2 Reduction via finite discretization

The optimization problem $(\mathcal{P})$ is challenging, since it involves an uncountable number of constraints. To address this difficulty, a natural approach is to consider a proxy of problem $(\mathcal{P})$, based on a finite discretization step. To describe our observation, for any positive integer $L$, let $\mathcal{Q}_L$ be the uniform grid of $2L + 1$ points on $[-M, M]$, that is, $\mathcal{Q}_L = \{\ell M / L\}_{\ell=-L}^{L}$. Denote by $\mathcal{G}_L$ the set of measurable functions from $\mathbb{R}^d \times \mathcal{S}$ to $\mathcal{Q}_L$. The fair optimal discretized predictor $g_L^* : \mathbb{R}^d \times \mathcal{S} \to \mathcal{Q}_L$ is defined as

$$g_L^* \in \arg\min_{g \in \mathcal{G}_L} \left\{ \mathcal{R}(g) \,:\, \max_{q \in \mathcal{Q}_L} \mathcal{U}\left(g, \{q\}\right) = 0 \right\} \ . \tag{$\mathcal{P}'_L$}$$

Note that unlike $f^*$, which takes values in the whole interval $[-M, M]$, the function $g_L^*$ only takes values in the uniform grid $\mathcal{Q}_L$.

The following lemma confirms the intuition that for large values of $L$, the risk of $g_L^*$ should be similar to that of $f^*$.

**Lemma 2.4.** *Let $\sigma^2 = \mathrm{Var}(Y)$. For every positive integer $L$, all solutions $g_L^*$ of $(\mathcal{P}_L')$ are fair in the sense of Definition 2.1. Moreover,*

$$\mathcal{R}(g_L^*) \leq \mathcal{R}(f^*) + 2\sigma\frac{M}{L} + \frac{M^2}{L^2} \ .$$

Interestingly, problem $(\mathcal{P}_L')$ can be solved analytically under the following mild assumption.

**Assumption 2.5.** *Assume, for all $s \in \mathcal{S}$, that the mappings $t \mapsto \mathbb{P}\left(\eta(X, s) \leq t \mid S = s\right)$ are continuous.*

It is satisfied if the random variable $\eta(X, S)$ does not have atoms conditionally on $S = \pm 1$.

**Proposition 2.6** (Optimal fair predictor)**.** *Under Assumption 2.5 for all positive integers $L$ a solution $g_L^*$ of problem $(\mathcal{P}_L')$ is given for all $(x, s) \in \mathbb{R}^d \times \mathcal{S}$ by*

$$g_L^*(x, s) = \underset{\ell \in \{-L,\dots,L\}}{\arg\min} \left\{-s\lambda_\ell^* + p_s\left(\eta(x, s) - \frac{\ell M}{L}\right)^2\right\} \times \frac{M}{L} \ , \tag{2}$$

*where $\lambda_{-L}^*, \dots, \lambda_L^*$ are solutions of*

$$\min_{\lambda \in \mathbb{R}^{2L+1}} \sum_{s \in \mathcal{S}} \mathbb{E}_{X \mid S=s} \max_\ell \left\{ s\lambda_\ell - p_s\left(\eta(X, s) - \frac{\ell M}{L}\right)^2 \right\} \ . \tag{3}$$

The proof of this result borrows ideas from [16, 18]. In particular, we first write problem $(\mathcal{P}_L')$ in the minmax form. It appears that its dual maxmin version can be solved analytically and Assumption 2.5 guarantees the strong duality.

The above result says that an optimal solution of the discretized fair regression problem $(\mathcal{P}_L')$ is obtained by first computing the standard regression function $\eta$ and then transforming this function via problems (2) and (3). In virtue of Lemma 2.4 a tempting approach to ultimately estimate the optimal fair regression function in problem $(\mathcal{P})$, would be to use an estimator of $g_L^*$, by first estimating the regression function $\eta$ and then implementing an empirical version of problem (3). The next section describe in more details this estimator and, crucially, justify its choice by proving non-asymptotic error bounds for its excess risk and fairness constraint.

## 3 Proposed approach

In what follows we propose a data-driven procedure $\hat{g}$, which is based on *two* data samples: a labeled sample $\mathcal{D}_n = (X_i, S_i, Y_i)_{i=1}^n \overset{\text{i.i.d.}}{\sim} \mathbb{P}$ of size $n$, and an independent unlabeled sample $\mathcal{D}_N' = (X_i', S_i')_{i=1}^N \overset{\text{i.i.d.}}{\sim} \mathbb{P}_{(X,S)}$ of size $N$, where $\mathbb{P}_{(X,S)}$ is the marginal distribution of $(X, S)$ induced by $\mathbb{P}$. That is, our algorithm is performed in a semi-supervised manner. The *principal goal* of this work is to construct a procedure $\hat{g}$ which meets two criteria:

$$\text{Fairness: } \mathbf{E}[\sup_{\mathcal{C} \in \mathbb{R}} \mathcal{U}(\hat{g}, \mathcal{C})] \leq \delta_{n,N}, \qquad \text{Risk optimality: } \mathbf{E}[\mathcal{E}(\hat{g})] \leq \delta_{n,N}',$$

where $\delta_{n,N}$ and $\delta_{n,N}'$ are two decreasing sequences of $n$ and $N$, the excess risk $\mathcal{E}(f) := \mathcal{R}(f) - \mathcal{R}(f^*)$, and $\mathbf{E}$ is the expectation taken *w.r.t.* the distribution of the observations $\mathcal{D}_n, \mathcal{D}_N'$.

The proposed method is a plug-in approach which mimics the conditions imposed on $g_L^*$ from Proposition 2.6. We require an off-the-shelf estimator $\hat{\eta}(X, S)$ of $\eta(X, S) = \mathbb{E}[Y \mid X, S]$ which is constructed using *only* the first *labeled* sample. This problem has been studied to a great extent and it is not of the main concern in this work. For instance such estimators include locally polynomial methods [39, 53], $k$-nearest neighbours [52, 20], random forests [9, 51], ridge and lasso regressions [3, 7], and many more. We also require the following, rather technical, assumption on the constructed estimator $\hat{\eta}$.

**Assumption 3.1.** *The mappings $t \mapsto \mathbb{P}\left(\hat{\eta}(X, s) \leq t \mid S = s\right)$ are almost surely continuous.*

We refer to [17] for an in-depth discussion on this assumption and an ad-hoc method which allows to satisfy this condition for any estimator $\hat{\eta}$ and any distribution $\mathbb{P}_{X \mid S=s}$ which admits a density *w.r.t.* the Lebesgue measure. Yet, this assumption is of little or no concern for the practitioner as we demonstrate in our experimental study in Section 4.

To proceed with our plug-in method, we first decompose the unlabeled sample $\mathcal{D}'_N$ into three groups $\mathcal{D}'_{N_{-1}}$, $\mathcal{D}'_{N_1}$ and $\mathcal{D}^S_N$ of sizes $N_{-1}, N_1$, and $N$ respectively. So that $\mathcal{D}'_{N_s} = \{X'_i : (X'_i, S'_i) \in \mathcal{D}'_N, S'_i = s\}$ for all $s \in \{-1, 1\}$ and $\mathcal{D}^S_N = \{S'_i : (X'_i, S'_i) \in \mathcal{D}'_N\}$.

Our next goal is to mimic the condition on $\lambda^*_{-L}, \ldots, \lambda^*_L$ imposed by Eq. (3), which requires the knowledge of $\eta$, $p_s$, and $\mathbb{P}_{X|S=s}$ for $s \in \mathcal{S}$ and $\ell \in \{-L, \ldots, L\}$. The estimator $\hat{p}_1$ of $p_1 = \mathbb{P}(S = 1)$ is based on the empirical frequencies on $\mathcal{D}^S_N$ and $\hat{p}_{-1} = 1 - \hat{p}_1$. For each $s \in \mathcal{S}$, the conditional expectation $\mathbb{E}_{X|S=s}$ is estimated using its empirical version on $\mathcal{D}'_{N_s}$ as $\hat{\mathbb{P}}_{X|S=s} = \frac{1}{N_s} \sum_{X' \in \mathcal{D}'_{N_s}} \delta_{X'}$. The *final estimator* $\hat{g}_L$ is then defined for all $(x, s) \in \mathbb{R}^d \times \mathcal{S}$ as

$$\hat{g}_L(x, s) = \underset{\ell \in \{-L, \ldots, L\}}{\arg\min} \left\{ -s\hat{\lambda}_\ell + \hat{p}_s \left( \hat{\eta}(x, s) - \frac{\ell M}{L} \right)^2 \right\} \times \frac{M}{L} \ , \tag{4}$$

where $\hat{\lambda}_{-L}, \ldots, \hat{\lambda}_L$ are solutions of

$$\min_{\lambda \in \mathbb{R}^{2L+1}} \sum_{s \in \mathcal{S}} \hat{\mathbb{E}}_{X|S=s} \max_\ell \left\{ s\lambda_\ell - \hat{p}_s \left( \hat{\eta}(X, s) - \frac{\ell M}{L} \right)^2 \right\} \ . \tag{5}$$

If the $\arg\min$ in Eq. (4) is not a singleton, we use the convention that the smallest value of $\ell$ is taken. Also notice that the minimization problem in Eq. (5) is convex. Therefore, it can be efficiently solved. In Section 3.2 we address this point and propose an efficient iterative algorithm based on the smoothing technique of Nesterov [45].

In summary, the proposed procedure is composed of two steps. First, we estimate the regression function $\eta$ by standard methods using only labeled data, and then we estimate the thresholds $\lambda^*_{-L}, \ldots, \lambda^*_L$ using *unlabeled* data and the estimator $\hat{\eta}$ constructed on the first step. In many applications of fairness, an accurate initial estimator $\hat{\eta}$ is already available. Thus, our work suggests that in order to transform $\hat{\eta}$ into a fair predictor it is sufficient to gather only *unlabeled* data and solve the minimization problem in Eq. (5), which may be much cheaper than training a fair predictor from scratch.

## 3.1  Rates of convergence

In this section we present the rates of convergence of the proposed algorithm for an arbitrary value of $L \in \mathbb{N}$. These bounds demonstrate a bias-variance trade-off and a way to select $L$ which optimizes it. We begin with bound on the violation of the fairness constraint of the proposed algorithm.

**Theorem 3.2.** *Under Assumption 3.1, there exists a universal constant $C > 0$ such that for each $L \in \mathbb{N}$ the proposed procedure $\hat{g}_L$ satisfies*

$$\mathbf{E}\left[ \sup_{\mathcal{C} \subset \mathbb{R}} \mathcal{U}(\hat{g}_L, \mathcal{C}) \right] \leq C \sum_{s \in \mathcal{S}} \sqrt{\frac{L}{p_s N}} \ .$$

*Proof sketch.* We first derive the first order optimality condition for the problem in Eq. (5). Since this problem is non-smooth (due to the $\max$) the optimality condition involves a sub-gradient of the objective. Using Assumption 3.1 we show that the non-smooth part of the objective has a little impact on the sub-gradient. On the final step, we show that the quantity of interest is controlled by a properly chosen empirical process plus the impact of the non-smooth part of the objective. $\square$

The bound depends only on the size of the *unlabeled* dataset, and not on the quality of the initial estimator $\hat{\eta}$. It can be intuitively explained by the fact that the notion of fairness in Definition 2.1 depends only on the conditional distribution of $X$ given $S$ and not on the regression function $\eta$. A consequence of our findings is that when a large unlabeled dataset is available, achieving fairness becomes an easy task based only on the recalibration step we propose.

The next bound is on the excess-risk of the proposed algorithm. It establishes the trade-off introduced by the discretization step.

**Theorem 3.3.** *Let Assumptions 2.5 and 3.1 be satisfied. Then there exists a universal constant $C > 0$ such that for all $L \in \mathbb{N}$, the proposed procedure $\hat{g}_L$ satisfies*

$$\mathbf{E}[\mathcal{E}(\hat{g}_L)] \leq CM^2 \sum_{s \in \mathcal{S}} \left( \sqrt{\frac{L^2}{p_s N}} + \frac{1}{2L} \right) + 8M\mathbf{E} \left\| \eta - \hat{\eta} \right\|_1 \ .$$

---

**Algorithm 1** Smoothed accelerated gradient descent

---
**Input:** temperature parameter $\beta$, number of iterations $T$
Initialize $\lambda_1 = z_1 = \tau_0 = 0$.
**for** $t = 1$ **to** $T$ **do**
$$\gamma_t = \frac{1-\tau_{t-1}}{\tau_t}, \tau_t = \frac{1+\sqrt{1+4\tau_{t-1}^2}}{2}$$
$$(\lambda_{t+1})_\ell = (z_t)_\ell - \frac{\beta}{2}\sum_{s\in\mathcal{S}}s\hat{\mathbb{E}}_{X|S=s}\left[\sigma_\beta\left(s(z_t)_\ell - \hat{p}_s\left(\hat{\eta}(X,s)-\ell M/L\right)^2\right)\right], \quad \forall\ell$$
$$z_{t+1} = (1-\gamma_t)\lambda_{t+1} + \gamma_t\lambda_t$$
**end for**
**Output:** $\lambda_T$

---

*Proof sketch.* The proof of this result goes in two steps. On the first step we leverage the form of the optimal predictor $g_L^*$ and the constructed plug-in rule $\hat{g}_L$ to show that $\mathcal{R}(\hat{g}_L) - \mathcal{R}(g_L^*)$ can be bounded by two terms. The first term involves the violation of the fairness constraints and is controlled by Theorem 3.2. The second term can be controlled by the estimation error of $\hat{\eta}$ and $\hat{p}_s$. Finally, we combine Lemma 2.4 with the bound on $\mathcal{R}(\hat{g}_L) - \mathcal{R}(g_L^*)$ to obtain the result on $\mathcal{E}(\hat{g}_L)$. □

Unlike the bound on fairness, the excess-risk bound already depends on the quality of $\hat{\eta}$. Importantly, the last term in the above bound decreases with $n$ instead of $p_s n$, that is, this term is not affected by the unbalanced distributions. Finally, from the excess-risk bound we can observe that the parameter $L$ should be chosen in an optimal way, performing the bias-variance trade-off. Setting $L = N^{1/4}$ in the previous results we immediately get the following corollary.

**Corollary 3.4.** *Let Assumptions 2.5 and 3.1 be satisfied and let $L = N^{1/4}$. Then there exists universal constants $C, C' > 0$ such that the proposed procedure $\hat{g}_L$ satisfies*

$$\mathbf{E}\left[\sup_{\mathcal{C}\subset\mathbb{R}}\mathcal{U}(\hat{g}_L,\mathcal{C})\right]\leq C\sum_{s\in\mathcal{S}}(p_s^{\frac{8}{6}}N)^{-\frac{3}{8}} \quad and \quad \mathbf{E}[\mathcal{E}(\hat{g}_L)]\leq C'M^2\sum_{s\in\mathcal{S}}(p_s^2 N)^{-\frac{1}{4}} + 8M\mathbf{E}\left\|\eta-\hat{\eta}\right\|_1 .$$

Note that the choice of $L$ is independent from the size of the labeled data $n$ and it does not affect the second term on the right hand side of the excess-risk guarantee. A careful analysis of our proof reveals that a data driven choice of $L$ that depends on $\hat{p}_s$ would improve the above result. Namely, instead of $p_s^2 N$ we could obtain $p_s N$. However, this proof is much more technical and is thus omitted. Finally, we remark that the result of Corollary 3.4 and both Theorems 3.2–3.3 explicitly assumes that $\hat{\lambda}_{-L},\ldots,\hat{\lambda}_L$ is a global minimizer of the problem in Eq. (5). In the next section we provide an optimization algorithm which finds an $\epsilon$ solution of the problem in Eq. (5).

### 3.2 Optimization algorithm

Recall that the proposed estimator sets $\hat{\lambda}_{-L},\ldots,\hat{\lambda}_L$ to be a solution of the minimization problem in Eq. (5). This problem is convex but non-smooth, thus subgradient methods can be used to numerically approximate a solution. While being optimal in a black-box optimization paradigm [46], subgradient methods often can be significantly accelerated if the structure of the non-smooth problem is "simple". In our setting, we follow the smoothing technique due to Nesterov [45], which leads to Algorithm 1. The key insight in this approach is to approximate the inner maximum in the objective function of Eq. (5) by a smooth convex function with Lipschitz gradient. This results in the LogSumExp (also known as soft-max) instead of the "hard" max. Such smoothed problem is then solved using an optimal method, such as the accelerated gradient descent [44].

To understand the proposed optimization algorithm, let us introduce some notation. For any vector $\lambda\in\mathbb{R}^{2L+1}$, the soft argmax (also known as Gibbs distribution) of $\lambda$ with the temperature parameter $\beta$ is defined component-wise for all $\ell\in\{-L,\ldots,L\}$ as $\sigma_\beta(\lambda)_\ell := \exp\left(\frac{\lambda_\ell}{\beta}\right)/\sum_{\ell=-L}^{L}\exp\left(\frac{\lambda_\ell}{\beta}\right)$. Finally, denote by $G:\mathbb{R}^{2L+1}\to\mathbb{R}$ the objective function of the minimization in Eq. (5). That is, the vector $\hat{\lambda} = (\hat{\lambda}_{-L},\ldots,\hat{\lambda}_L)^\top$ is a minimizer of $\min_{\lambda\in\mathbb{R}^{2L+1}}G(\lambda)$. To find an $\varepsilon$-solution of this problem we run Algorithm 1, which takes as an input two parameters $T\in\mathbb{N}$ and $\beta > 0$.

| Method | CRIME MSE | CRIME KS | LAW MSE | LAW KS | NLSY MSE | NLSY KS | STUD MSE | STUD KS | UNIV MSE | UNIV KS |
|---|---|---|---|---|---|---|---|---|---|---|
| RLS | .033±.003 | .55±.06 | .107±.010 | .15±.02 | .153±.016 | .73±.07 | 4.77±.49 | .50±.05 | 2.24±.22 | .14±.01 |
| RLS+Berk | .037±.004 | .16±.02 | .121±.013 | .10±.01 | .189±.019 | .49±.05 | 5.28±.57 | .32±.03 | 2.43±.23 | .05±.01 |
| RLS+Oneto | .037±.004 | .14±.01 | .112±.012 | .07±.01 | .156±.016 | .50±.05 | 5.02±.54 | .23±.02 | 2.44±.26 | .05±.01 |
| RLS+Ours | .035±.003 | .22±.02 | .117±.011 | .04±.01 | .177±.017 | .16±.02 | 5.14±.46 | .11±.01 | 2.63±.24 | .04±.01 |
| KRLS | .024±.003 | .52±.05 | .040±.004 | .09±.01 | .061±.006 | .58±.06 | 3.85±.36 | .47±.05 | 1.43±.15 | .10±.01 |
| KRLS+Oneto | .028±.003 | .19±.02 | .046±.004 | .05±.01 | .066±.007 | .06±.01 | 4.07±.39 | .18±.02 | 1.46±.13 | .04±.01 |
| KRLS+Perez | .033±.003 | .25±.02 | .048±.005 | .04±.01 | .065±.007 | .08±.01 | 3.97±.38 | .14±.02 | 1.50±.15 | .06±.01 |
| KRLS+Ours | .032±.003 | .12±.01 | .050±.005 | .02±.01 | .070±.007 | .05±.01 | 4.04±.37 | .06±.01 | 1.51±.15 | .02±.01 |
| RF | .020±.002 | .45±.04 | .046±.005 | .11±.01 | .055±.006 | .55±.06 | 3.59±.39 | .45±.05 | 1.31±.13 | .10±.01 |
| RF+Raff | .030±.003 | .21±.02 | .058±.006 | .06±.01 | .066±.006 | .08±.01 | 4.28±.40 | .09±.01 | 1.38±.12 | .02±.01 |
| RF+Agar | .029±.003 | .13±.01 | .050±.005 | .04±.01 | .065±.006 | .07±.01 | 3.87±.41 | .07±.01 | 1.40±.13 | .02±.01 |
| RF+Ours | .031±.003 | .09±.01 | .061±.006 | .03±.01 | .064±.007 | .05±.01 | 3.93±.36 | .05±.01 | 1.41±.14 | .02±.01 |

Table 1: Results for all the datasets and all the methods concerning MSE and KS when the sensitive feature is exploited in the functional form of the model.

**Theorem 3.5.** *For every $L > 0$ and every $\varepsilon > 0$ the output $\lambda_T$ of Algorithm 1 with*

$$\beta = \frac{M^2\sqrt{2L+1}}{T \log(2L+1)} \quad and \quad T = \left\lceil \frac{256M^2}{\varepsilon}\sqrt{(2L+1)}\log(2L+1) \right\rceil ,$$

*satisfies $G(\lambda_T) - G(\hat{\lambda}) \leq \varepsilon$.*

Unlike subgradient methods that require $T = O(\varepsilon^{-2})$ iterations to achieve $\varepsilon$-solution, smoothing technique allows to achieve $T$ of order $\varepsilon^{-1}$ as stated in Theorem 3.5. More precisely, when we set $L = N^{1/4}$ as suggested by Corollary 3.4, $T = O(\varepsilon^{-1}N^{1/8}\log(N))$. Following our statistical results a reasonable choice of the optimization accuracy is $\varepsilon = O(N^{-1/4})$, implying that the total amount of iterations $T = O(N^{3/8}\log(N))$.

**Remark 3.6.** *We did not attempt to improve the constant 256 present in the choice of $T$, as our main interest in this result is the dependence on $N$ and $\varepsilon$.*

On each iteration, Algorithm 1 computes the soft argmax function and averages it over the unlabeled dataset, which can be done in time linear in $N$. Note that the averaging step involves only unlabeled data and can be pre-computed before running the algorithm. Finally, to compute the estimator $\hat{g}_L(x)$ at a new point $x$ (see Eq. (4)) we need to find the minimum over a finite set, which is performed in time linear in $L = N^{1/4}$.

## 4 Empirical study

In this section, we present numerical experiments[3] with the proposed fair regression estimator.

**Experimental setting.** In all experiments, we collect statistics on the test set $\mathcal{T} = \{(X_i, S_i, Y_i)\}_{i=1}^{|\mathcal{T}|}$. The empirical mean squared error (MSE) is defined as

$$\text{MSE}(g) = \frac{1}{|\mathcal{T}|} \sum_{(X,S,Y) \in \mathcal{T}} (Y - g(X,S))^2 .$$

We also would like to measure the violation of fairness constraint imposed by Definition 2.1. Since it involves a computationally expensive TV variation distance, we replace it by the empirical Kolmogorov-Smirnov distance:

$$\text{KS}(g) = \sup_{t \in \mathbb{R}} \left| \frac{1}{|\mathcal{T}_{-1}|} \sum_{(X,Y) \in \mathcal{T}_{-1}} \mathbf{1}_{\{g(X,-1) \leq t\}} - \frac{1}{|\mathcal{T}_{+1}|} \sum_{(X,Y) \in \mathcal{T}_{+1}} \mathbf{1}_{\{g(X,+1) \leq t\}} \right| ,$$

where for all $s \in \{-1, +1\}$ the set $\mathcal{T}_s = \{(X, S, Y) \in \mathcal{T} : S = s\}$. For all datasets we split the data in two parts (70% train and 30% test), this procedure is repeated 30 times, and we report the average performance on the test set alongside its standard deviation. We employ the 2-steps 10-fold CV procedure considered by [22] to select the best hyperparameters with the training set. In the first step, we shortlist all the hyperparameters with MSE close to the best one (in our case, the hyperparameters which lead to 10% larger MSE w.r.t. the best MSE). Then, from this list, we select the hyperparameters with the lowest KS.

**Methods.** We compare our method to different fair regression approaches for both linear and non-linear regression. In the case of linear models we consider the following methods: Linear RLS plus [6] (RLS+Berk), Linear RLS plus [47] (RLS+Oneto), and Linear RLS plus Our Method (RLS+Ours), where RLS is the abbreviation of Regularized Least Squares. In the case of non-linear models we compare to the following methods: Kernel RLS (KRLS), Kernel RLS plus [47] (KRLS+Oneto), Kernel RLS plus [48] (KRLS+Perez), Kernel RLS plus Our Method (KRLS+Ours), Random Forests (RF), Random Forests plus [49] (RF+Raff), Random Forests plus [2][4] (RF+Agar), and Random Forests plus Our Method (RF+Ours).

The hyperparameters of the methods are set as follows. As our theory suggests that $L = N^{1/4}$ leads to a statistically grounded approach, we choose $L \in \{6, 12, 24\}$ since the size of the considered datasets is smaller than $24^4 \approx 3 \times 10^5$ and $\beta \in \{0.1, 0.01\}$. For RLS we set the regularization hyperparameters $\lambda \in 10^{\{-4.5, -3.5, \cdots, 3\}}$ and for KRLS we set $\lambda \in 10^{\{-4.5, -3.5, \cdots, 3\}}$ and $\gamma \in 10^{\{-4.5, -3.5, \cdots, 3\}}$. Finally, for RF we set to 1000 the number of trees and for the number of features to select during the tree creation we search in $\{d^{1/4}, d^{1/2}, d^{3/4}\}$.

**Datasets.** In order to analyze the performance of our methods and test it against the state-of-the-art alternatives, we consider five benchmark datasets, CRIME, LAW, NLSY, STUD, and UNIV, which are briefly described below:

*Communities&Crime (CRIME)* contains socio-economic, law enforcement, and crime data about communities in the US [50] with 1994 examples. The task is to predict the number of violent crimes per $10^5$ population (normalized to $[0, 1]$) with race as the protected attribute. Following [12], we made a binary sensitive attribute $s$ as to the percentage of black population, which yielded 970 instances of $s = 1$ with a mean crime rate 0.35 and 1024 instances of $s = -1$ with a mean crime rate 0.13.

*Law School (LAW)* refers to the Law School Admissions Councils National Longitudinal Bar Passage Study [56] and has 20649 examples. The task is to predict a students GPA (normalized to $[0, 1]$) with race as the protected attribute (white versus non-white).

*National Longitudinal Survey of Youth (NLSY)* involves survey results by the U.S. Bureau of Labor Statistics that is intended to gather information on the labor market activities and other life events of several groups [10]. Analogously to [37] we model a virtual company's hiring decision assuming that the company does not have access to the applicants' academic scores. We set as target the person's GPA (normalized to $[0, 1]$), with race as sensitive attribute.

*Student Performance (STUD)*, approaches 649 students achievement (final grade) in secondary education of two Portuguese schools using 33 attributes [19], with gender as the protected attribute.

*University (UNIV)*[5] is a proprietary and highly sensitive dataset containing all the data about the past and present students enrolled at the University of *Genoa*. In this study we take into consideration students who enrolled, in the academic year 2017-2018. The dataset contains 5000 instances, each one described by 35 attributes (both numeric and categorical) about ethnicity, gender, financial status, and previous school experience. The scope is to predict the average grades at the end of the first semester, with gender as the protected attribute.

**Comparison w.r.t. state-of-the-art.** In Table 1, we present the performance of different methods on various datasets described in Section 4. Our findings indicate that the proposed method is generally superior or competitive with state-of-the-art methods. In particular, our method is extremely good in enforcing fairness, even though, often, this comes at the cost of a slight increase in the MSE. Overall, RF+Ours tends to be the most effective method, and the one we would recommend to use in practice.

## 5   Discussion and conclusion

We proposed a new method to fair regression, which is able to estimate the optimal fair regression function, when the demographic parity constraint is imposed. This approach is very general and can be employed on top of any standard estimator, by means of the recalibration step which only involves an additional independent unlabeled dataset. This step can be efficiently implemented by solving a small-scale convex optimization problem. We derived non-asymptotic error rates for this estimator, relative to both the squared risk and a fairness violation based on the total variation distance. Numerical experiments demonstrated that the proposed method is effective and often superior to previous fair regression methods. In future it would be valuable to study the theoretical impact of the smoothing parameter on the risk/fairness trade-off as well as to understand the optimality of our bounds. Finally, an important open problem is whether an estimator having the same guarantees as the proposed one, could be constructed on the basis of a single dataset, used both to estimate the regression function and the recalibrations step.

## Broader impact

Although theoretical, our work may have at least two indirect positive future societal effects. First, the proposed algorithm is easy to implement and use, since it works in a post-processing regime. This makes it potentially attractive to practitioners who use computationally demanding methods. Second, our procedure comes with strong theoretical guarantees, making our contribution more reliable in practice.
Despite this potentially positive impact, one should be aware that the notion of demographic parity is only a way to formalize the idea of fairness in a rigorous mathematical manner and, as other similar formalizations, it might not reflect the reality. Hence, while this fact does not compromise our theoretical contribution, one has to pay extra care to the choice of the fairness notion when machine learning algorithms are to be deployed to society.

## Acknowledgments and Disclosure of Funding

This work was partially supported Amazon Web Services and SAP SE.

## Footnotes

*evgenii.chzhen@universite-paris-saclay.fr

[2]For simplicity, in what follows we only consider the case of a binary sensitive feature. However, our methodology extends to non-binary case.

[3]The source of our method can be found at `https://github.com/lucaoneto/NIPS2020_Fairness`.

[4]We thank the authors for sharing a prototype of their code.

[5]The data and the research are related to the project DROP@UNIGE of the University of Genoa.

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
