[Supplementary Material]

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

[6]Theorem 3.3 provides a bound on $\mathcal{E}(\hat{g}) = \mathcal{R}(\hat{g}) - \mathcal{R}(f^*)$, while Theorem C.5 is stated on $\mathcal{R}(\hat{g}) - \mathcal{R}(g^*)$. The result of Theorem 3.3 is recovered immediately from Lemma 2.4 and Theorem C.5.

[7]The reader should not confuse $\eta$ defined in the main body with $\eta$ defined in this section.

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

# Supplementary Material

Below we give an overview of the structure of the supplementary material and highlight the main novel results of this work.

- Appendix A is mainly devoted to the derivation of the expression for the optimal predictor $g_L^*$. The proof of Lemma 2.4 is also placed in this section.
- Appendix B states general preparation results which are used for the proof of fairness rates. Appendix B.1 is devoted to the proof of Theorem 3.2, which establishes fairness guarantees of the proposed procedure.
- Similarly, Appendix C starts by stating supporting results, whose proofs are postponed to Section C.2. Appendix C.1 is devoted to the proof of Theorem 3.3, which establishes guarantees on the excess-risk of the proposed procedure.
- Appendix D is devoted to the optimization part of our contribution and establishes guarantees on Algorithm 1.
- Appendix F shows the impact of unlabeled data on the performance of the estimator.

Let us also mention that in the supplementary material we omit the underscript $L$, when no confusion can rise. That is, instead of $g_L^*$ and $\hat{g}_L$ we write $g^*$ and $\hat{g}$ respectively. Finally, before proceeding further, let us point out one technical subtlety: in what follows it is assumed that the estimator $|\hat{\eta}(\cdot, s)| \leq M$, this assumption is never restrictive in practice as long as $M$ is known. Indeed, if $\hat{\eta}(\cdot, s)$ take values outside of $[-M, M]$, then its truncationn on this interval is strictly better in terms of the $\ell_1$ error, since the true $|\eta(\cdot, s)| \leq M$.

## A  Derivation of the optimal predictor and its properties

First we state a rather intuitive statement. Informally, if the signal $Y$ is almost surely bounded on the interval $[-M, M]$, then the fair optimal predictor $f^*$ is also bounded almost surely on the interval $[-M, M]$. This result allows to consider only those predictors $f$, which take values in $[-M, M]$.

**Lemma A.1.** *Assume that $|Y| \leq M$ almost surely, then $|f^*(X, S)| \leq M$ almost surely.*

*Proof.* Let $f^*$ be the minimizer of problem $\mathcal{P}$. Denote by $f \mapsto \Pi_f$ the projection defined as

$$\Pi_f(x, s) = f(x, s)\mathbf{1}_{\{|f(x,s)| \leq M\}} + M \operatorname{sign}(f(x, s))\mathbf{1}_{\{|f(x,s)| > M\}} \ .$$

Now our goal is to show that $\Pi_{f^*}$ is fair in the sense of Definition 2.1 and that its risk is upper bounded by the risk of $f^*$. This would imply that $\Pi_{f^*} = f^*$ almost surely. The fairness of $\Pi_{f^*}$ follows directly from the fairness of $f^*$. Moreover, we can write

$$
\begin{aligned}
\mathbb{E}(Y - \Pi_{f^*}(X, S))^2 &= \mathbb{E}(Y - f^*(X, S) + f^*(X, S) - \Pi_{f^*}(X, S))^2 \\
&= \mathbb{E}(Y - f^*(X, S))^2 + 2\mathbb{E}(Y - f^*(X, S))(f^*(X, S) - \Pi_{f^*}(X, S)) \\
&\quad + \mathbb{E}(f^*(X, S) - \Pi_{f^*}(X, S))^2 \ .
\end{aligned}
$$

Let us introduce the following notation

$$Z = 2(Y - f^*(X, S))(f^*(X, S) - \Pi_{f^*}(X, S)) + (f^*(X, S) - \Pi_{f^*}(X, S))^2 \ .$$

Notice that

$$Z = (2Y - f^*(X, S) - \Pi_{f^*}(X, S))(f^*(X, S) - \Pi_{f^*}(X, S)) \ .$$

If we can show that $Z \leq 0$ almost surely, the proof is finished. To see this, we first notice that

$$
\begin{aligned}
f^*(X, S) - \Pi_{f^*}(X, S) &= (|f^*(X, S)| - M) \operatorname{sign}(f^*(X, S))\mathbf{1}_{\{|f^*(X,S)| > M\}} \ , \\
f^*(X, S) + \Pi_{f^*}(X, S) &= 2f^*(X, S)\mathbf{1}_{\{|f^*(x,s)| \leq M\}} \\
&\quad + (M + |f^*(X, S)|) \operatorname{sign}(f^*(X, S))\mathbf{1}_{\{|f^*(X,S)| > M\}} \ .
\end{aligned}
$$

After simple algebraic manipulations $Z$ can be expressed as

$$Z = (2Y \operatorname{sign}(f^*(X,S)) - M - |f^*(X,S)|) \, (|f^*(X,S)| - M) \, \mathbf{1}_{\{|f^*(X,S)|>M\}}$$
$$\leq 2 \, (Y \operatorname{sign}(f^*(X,S)) - M) \, (|f^*(X,S)| - M) \, \mathbf{1}_{\{|f^*(X,S)|>M\}}$$
$$\leq 2 \, (|Y| - M) \, (|f^*(X,S)| - M) \, \mathbf{1}_{\{|f^*(X,S)|>M\}} \ .$$

Finally, since $|Y| \leq M$ we conclude. □

Now, we prove Lemma 2.4, which gives a theoretical justification to the reduction scheme and the introduction of $g_L^*$. Let us recall the statement of this result first.

**Lemma** (Lemma 2.4). *For every positive integer L, all solutions $g_L^*$ of $(\mathcal{P}_L')$ are fair in the sense of Definition 2.1. Moreover*

$$\mathcal{R}(g_L^*) \leq \mathcal{R}(f^*) + 2\sigma \frac{M}{L} + \frac{M^2}{L^2} \ ,$$

*where $\sigma^2 = \operatorname{Var}(Y)$.*

*Proof of Lemma 2.4.* First we show that $g_L^*$ is fair. Fix arbitrary $C \in [-M, M]$, thus for any $s \in \mathcal{S}$ we can write

$$\mathbb{P}\left(g_L^*(X,S) \in C \mid S = s\right) = \mathbb{P}\left(g_L^*(X,S) \in C \cap \mathcal{Q}_{L,M} \mid S = s\right)$$
$$= \sum_{y \in C \cap \mathcal{Q}_{L,M}} \mathbb{P}\left(g_L^*(X,S) = y \mid S = s\right) \ .$$

Every $y \in C \cap \mathcal{Q}_{L,M}$ can be expressed as $\ell M/L$ for some $\ell \in \{-L, \ldots, L\}$ and for every $\ell \in \{-L, \ldots, L\}$

$$\mathbb{P}\left(g_L^*(X,S) = \ell M/L \mid S = -1\right) = \mathbb{P}\left(g_L^*(X,S) = \ell M/L \mid S = 1\right) \ ,$$

which implies that $g_L^*$ is fair.

Finally, to demonstrate the inequality in this result we first construct an operator $T_L : \mathcal{F} \to \mathcal{G}_{L,M}$ defined point-wise for all $(x, s) \in \mathbb{R}^d \times \mathcal{S}$ as

$$(T_L(f))(x,s) = \lfloor Lf(x,s)/M \rfloor M/L \ ,$$

where for $x \in \mathbb{R}$, $\lfloor x \rfloor$ stands for the closest integer smaller or equal to $x$. Now, we show that $T_L(f^*)$ is feasible for problem $(\mathcal{P}_L')$. Indeed, for any $\ell \in \{-L, \ldots, L-1\}$ and any $(x,s) \in \mathbb{R}^d \times \mathcal{S}$, by construction of $T_L$, we have

$$(T_L(f^*))(x,s) = \ell M/L \qquad \Leftrightarrow \qquad f^*(x,s) \in \left[\frac{\ell M}{L}, \frac{(\ell+1)M}{L}\right) \ .$$

Therefore, since $f^*$ is fair and the set $[\ell M/L, (\ell+1)M/L)$ is Borel we have for all $\ell \in \{-L, \ldots, L-1\}$

$$\mathbb{P}\left((T_L(f^*))(X,S) = \ell M/L \mid S = -1\right) = \mathbb{P}\left((T_L(f^*))(X,S) = \ell M/L \mid S = 1\right) \ .$$

Moreover, we also have for all $(x,s) \in \mathbb{R}^d \times \mathcal{S}$

$$T_L(f^*)(x,s) = M \qquad \Leftrightarrow \qquad f^*(x,s) = M \ ,$$

which implies that for $\ell = L$ we have

$$\mathbb{P}\left((T_L(f^*))(X,S) = \ell M/L \mid S = -1\right) = \mathbb{P}\left((T_L(f^*))(X,S) = \ell M/L \mid S = 1\right) \ .$$

Thus, $T_L(f^*)$ is feasible for problem $(\mathcal{P}_L')$ and we can write

$$\mathbb{E}(Y - g_L^*(X,S))^2 \leq \mathbb{E}\left(Y - (T_L(f^*))(X,S)\right)^2$$
$$= \mathbb{E}(Y - f^*(X,S))^2 + \mathbb{E}\left(f^*(X,S) - (T_L(f^*))(X,S)\right)^2$$
$$+ 2\mathbb{E}(Y - f^*(X,S))(f^*(X,S) - (T_L(f^*))(X,S)) \ .$$

Notice that for all $(x, s)$ we have $|f^*(x, s) - (T_L(f^*))(x, s)| \leq M/L$, and thus using the Cauchy-Schwartz inequality we get

$$\mathbb{E}(Y - g_L^*(X, S))^2 \leq \mathbb{E}(Y - f^*(X, S))^2 + 2M \frac{\sqrt{\mathbb{E}(Y - f^*(X, S))^2}}{L} + \frac{M^2}{L^2} \ .$$

Finally, since $f(x, s) \equiv \mathbb{E}[Y]$ is a feasible function for problem $(\mathcal{P})$, we have

$$\mathbb{E}(Y - f^*(X, S))^2 \leq \mathrm{Var}(Y) \ ,$$

which concludes the proof. $\qquad\qquad\square$

The next proof is devoted to the derivation of the optimal predictor $g_L^*$ provided in Proposition 2.6. Below we recall the statement of Proposition 2.6.

**Proposition** (Proposition 2.6). *Under Assumption 2.5 for all positive integers $L$ a solution $g_L^*$ of problem $(\mathcal{P}'_L)$ is given for all $(x, s) \in \mathbb{R}^d \times \mathcal{S}$ by*

$$g_L^*(x, s) = \underset{\ell \in \{-L,\ldots,L\}}{\arg\min} \ \{-s\lambda_\ell^* + Z_\ell(x, s)\} \times \frac{M}{L} \ ,$$

*where, for every $s \in \mathcal{S}$ and $\ell \in \{-L, \ldots, L\}$, we have defined the quantity $Z_\ell(x, s) = p_s \left( \eta(X, s) - \frac{\ell M}{L} \right)^2$ and $\lambda_{-L}^*, \ldots, \lambda_L^*$ are solutions of*

$$\min_{\lambda \in \mathbb{R}^{2L+1}} \sum_{s \in \mathcal{S}} \mathbb{E}_{X|S=s} \max_\ell \{s\lambda_\ell - Z_\ell(X, s)\} \ .$$

*Proof of Proposition 2.6.* Our goal is to solve the following problem

$$\min_g \max_{\lambda \in \mathbb{R}^{2L+1}} \mathbb{E}(Y - g(X, S))^2$$

$$+ \sum_{\ell=-L}^{L} \lambda_\ell \left( \mathbb{P}\left( g(X, -1) = \ell M/L \,|\, S = -1 \right) - \mathbb{P}\left( g(X, 1) = \ell M/L \,|\, S = 1 \right) \right) \ .$$

First of all notice that the minimization of $\mathbb{E}(Y - g(X, S))^2$ is equivalent to the minimization of $\mathbb{E}_{(X,S)}(\eta(X, S) - g(X, S))^2$, where $\eta(X, S) = \mathbb{E}[Y|X, S]$. Therefore, instead of the above saddle point problem we target a solution of

$$\min_g \max_{\lambda \in \mathbb{R}^{2L+1}} \mathbb{E}_{(X,S)}(\eta(X, S) - g(X, S))^2$$

$$+ \sum_{\ell=-L}^{L} \lambda_\ell \left( \mathbb{P}\left( g(X, -1) = \ell M/L \,|\, S = -1 \right) - \mathbb{P}\left( g(X, 1) = \ell M/L \,|\, S = 1 \right) \right) \ .$$

Since $s \in \{-1, 1\}$, then the objective function of this saddle point problem can be rewritten as

$$\sum_{s \in \mathcal{S}} \left( p_s \mathbb{E}_{X|S=s}(\eta(X, s) - g(X, s))^2 - s \sum_{\ell=-L}^{L} \lambda_\ell \mathbb{P}\left( g(X, s) = \ell M/L \,|\, S = s \right) \right) \ ,$$

where $p_s = \mathbb{P}(S = s)$. Moreover, since $\sum_{\ell=-L}^{L} \mathbf{1}_{\{g(X,s)=\ell M/L\}} \equiv 1$ we can rewrite the original saddle point problem as

$$\min_g \max_{\lambda \in \mathbb{R}^{2L+1}} \sum_{s \in \mathcal{S}} \mathbb{E}_{X|S=s} \left[ \sum_{\ell=-L}^{L} \left( p_s(\eta(X, s) - \ell M/L)^2 - s\lambda_\ell \right) \mathbf{1}_{\{g(X,s)=\ell M/L\}} \right] \ .$$

Let us first solve the dual $\max\min$ problem, that is, we would like to find a solution of

$$\max_{\lambda \in \mathbb{R}^{2L+1}} \min_g \sum_{s \in \mathcal{S}} \mathbb{E}_{X|S=s} \left[ \sum_{\ell=-L}^{L} \left( p_s(\eta(X, s) - \ell M/L)^2 - s\lambda_\ell \right) \mathbf{1}_{\{g(X,s)=\ell M/L\}} \right] \ .$$

Clearly, for every fixed $\lambda \in \mathbb{R}^{2L+1}$ the solution of minimization problem inside is given by $\tilde{g}_\lambda$ defined point-wise as

$$\tilde{g}_\lambda(x, s) = \arg\min_\ell \left\{ p_s(\eta(X, s) - \ell M/L)^2 - s\lambda_\ell \right\} M/L \ .$$

Therefore, the $\max \min$ problem boils down to

$$\max_{\lambda \in \mathbb{R}^{2L+1}} \sum_{s \in \mathcal{S}} \mathbb{E}_{X|S=s} \left[ \min_\ell \left\{ p_s(\eta(X, s) - \ell M/L)^2 - s\lambda_\ell \right\} \right] \ .$$

Which is equivalent to

$$- \min_{\lambda \in \mathbb{R}^{2L+1}} \sum_{s \in \mathcal{S}} \mathbb{E}_{X|S=s} \left[ \max_\ell \left\{ s\lambda_\ell - p_s(\eta(X, s) - \ell M/L)^2 \right\} \right] \ .$$

As we are only interested in the minimizer of the above problem and not in the value of the minimum, we can write that the above problem is equivalent in this sense to

$$- \min_{\lambda \in \mathbb{R}^{2L+1}} \sum_{s \in \mathcal{S}} \mathbb{E}_{X|S=s} \left[ \max_\ell \left\{ s\lambda_\ell + 2p_s \frac{\ell M}{L} \eta(X, s) - p_s \frac{l^2 M^2}{L^2} \right\} \right] \ .$$

The objective function of the above minimization problem is convex and is uniformly lower-bounded. The convexity is obvious. Let us show that it is lower bounded. We have the following sequence

$$\sum_{s \in \mathcal{S}} \mathbb{E}_{X|S=s} \left[ \max_\ell \left\{ s\lambda_\ell + 2p_s \frac{\ell M}{L} \eta(X, s) - p_s \frac{l^2 M^2}{L^2} \right\} \right]$$

$$\geq \sum_{s \in \mathcal{S}} \max_\ell \left\{ \mathbb{E}_{X|S=s} \left[ s\lambda_\ell + 2p_s \frac{\ell M}{L} \eta(X, s) - p_s \frac{l^2 M^2}{L^2} \right] \right\}$$

$$= \sum_{s \in \mathcal{S}} \max_\ell \left\{ s\lambda_\ell + 2p_s \frac{\ell M}{L} \mathbb{E}_{X|S=s}[\eta(X, s)] - p_s \frac{l^2 M^2}{L^2} \right\}$$

$$\geq \max_\ell \left\{ \sum_{s \in \mathcal{S}} \left( s\lambda_\ell + 2p_s \frac{\ell M}{L} \mathbb{E}_{X|S=s}[\eta(X, s)] - p_s \frac{l^2 M^2}{L^2} \right) \right\}$$

$$= \max_\ell \left\{ 2\frac{\ell M}{L} \sum_{s \in \mathcal{S}} p_s \mathbb{E}_{X|S=s}[\eta(X, s)] - \frac{l^2 M^2}{L^2} \sum_{s \in \mathcal{S}} p_s \right\}$$

$$= \max_\ell \left\{ 2\frac{\ell M}{L} \mathbb{E}[Y] - \frac{l^2 M^2}{L^2} \right\}$$

$$= \max_\ell \left\{ \left( \mathbb{E}[Y] - \frac{\ell M}{L} \right)^2 \right\} - \mathbb{E}[Y]^2 \geq 0 \ .$$

To conclude the proof notice that under Assumption 2.5, the first order optimality condition for the minimization over $\lambda$ reads for all $\ell \in \{-L, \ldots, L\}$ as

$$\sum_{s \in \mathcal{S}} s\mathbb{P}_{X|S=s} \left( \tilde{g}_{\lambda^*}(X, s) = \frac{\ell M}{L} \right) = 0 \ ,$$

where $\lambda^*$ is a minimizer. Which implies that $\tilde{g}_{\lambda^*}$ is fair and thus is feasible for problem $(\mathcal{P}'_L)$. Using this argument, it is easy to see that $\mathcal{R}(\tilde{g}_{\lambda^*}) = \mathcal{R}(g^*)$ which concludes the proof. $\qquad\square$

The next proposition shows that the thresholds $\lambda^*_{-L}, \ldots, \lambda^*_L$ can be found in a compact region. Note that the same, line by line, proof can be applied for $\hat{\lambda}_{-L}, \ldots, \hat{\lambda}_L$, which is thus omitted.

**Proposition A.2.** *The minimization problem in Eq.* (3) *admits a global minimizer* $\lambda^*_{-L}, \ldots, \lambda^*_L$ *which satisfies*

$$\min_{\ell \in \{-L, \ldots, L\}} \{\lambda^*_\ell\} = 0, \qquad \max_{\ell \in \{-L, \ldots, L\}} \{\lambda^*_\ell\} \leq 4M^2 \ .$$

*Proof.* Before proceeding to the proof of this result let us first introduce some notation. We denote by $H(\lambda_{-L}, \ldots, \lambda_L)$ the objective function of the minimization problem in Eq. (3). That is,

$$H(\lambda_{-L}, \ldots, \lambda_L) = \sum_{s \in \mathcal{S}} \mathbb{E}_{X|S=s} \left[ \max_{\ell \in \{-L, \ldots, L\}} \left\{ s\lambda_\ell - p_s \left( \eta(X, s) - \frac{\ell M}{L} \right)^2 \right\} \right] \ .$$

Fix any minimizing sequence $\lambda^k = (\lambda_{-L}^k, \ldots, \lambda_L^k)^\top$ of $H(\lambda_{-L}, \ldots, \lambda_L)$, then for all $\varepsilon > 0$ there exists $K \in \mathbb{N}$ such that for all $k \geq K$ it holds that

$$H(0, \ldots, 0) + \varepsilon \geq H(\lambda_{-L}^k, \ldots, \lambda_L^k) \ .$$

Furthermore, notice that for any $(\lambda_\ell)_{\ell=-L, \ldots, L}$ and any $c \in \mathbb{R}$ it holds that

$$H(\lambda_{-L}, \ldots, \lambda_L) = H(\lambda_{-L}, \ldots, \lambda_L) + \sum_{s \in \mathcal{S}} sc$$

$$= \sum_{s \in \mathcal{S}} \mathbb{E}_{X|S=s} \left[ \max_{\ell \in \{-L, \ldots, L\}} \left\{ s(\lambda_\ell + c) - p_s \left( \eta(X, s) - \frac{\ell M}{L} \right)^2 \right\} \right]$$

$$= H(\lambda_{-L} + c, \ldots, \lambda_L + c) \ ,$$

which implies that $(\lambda_\ell)_{\ell=-L, \ldots, L} + c$ achieves the same value of the objective function. Thus, if we introduce $\tilde{\lambda}^k = (\tilde{\lambda}_{-L}^k, \ldots, \tilde{\lambda}_L^k)^\top$ such that for all $\ell \in \{-L, \ldots, L\}$ and all $k \in \mathbb{N}$ it holds that $\tilde{\lambda}_{-\ell}^k = \lambda_{-\ell}^k + c^k$ with $c^k = -\min_{\ell \in \{-L, \ldots, L\}} \lambda_{-\ell}^k$, then $\tilde{\lambda}^k$ is also a minimizing sequence. Hence, by the definition of a minimizing sequence, for all $\varepsilon > 0$ there exists $K \in \mathbb{N}$ such that for all $k \geq K$

$$H(0, \ldots, 0) + \varepsilon \geq H(\lambda_{-L}^k, \ldots, \lambda_L^k) = H(\tilde{\lambda}_{-L}^k, \ldots, \tilde{\lambda}_L^k) \ . \tag{6}$$

Using the definition of $H$ we can upperbound $H(0, \ldots, 0)$ as follows

$$H(0, \ldots, 0) = -\sum_{s \in \mathcal{S}} p_s \mathbb{E}_{X|S=s} \left[ \min_{\ell \in \{-L, \ldots, L\}} \left( \eta(X, s) - \frac{\ell M}{L} \right)^2 \right]$$

$$= -\mathbb{E}_{(X,S)} \left[ \min_{\ell \in \{-L, \ldots, L\}} \left( \eta(X, S) - \frac{\ell M}{L} \right)^2 \right] \leq 0 \ . \tag{7}$$

Substituting Eq. (7) to Eq. (6) we get for all $k \geq K$ that

$$\varepsilon \geq H(\tilde{\lambda}_{-L}^k, \ldots, \tilde{\lambda}_L^k) \ .$$

Moreover, for any $\lambda_{-L}, \ldots, \lambda_L$

$$H(\lambda_{-L}, \ldots, \lambda_L) = \sum_{s \in \mathcal{S}} \mathbb{E}_{X|S=s} \left[ \max_{\ell \in \{-L, \ldots, L\}} \left\{ s\lambda_\ell - p_s \left( \eta(X, s) - \frac{\ell M}{L} \right)^2 \right\} \right]$$

$$\geq \max\{\lambda_\ell\} - \min\{\lambda_\ell\} - 4M^2 \ , \tag{8}$$

where the inequality is obtained from the fact that for all $s \in \mathcal{S}$ it holds that

$$\max_{\ell \in \{-L, \ldots, L\}} \left\{ s\lambda_\ell - p_s \left( \eta(X, s) - \frac{\ell M}{L} \right)^2 \right\} \geq \max_{\ell \in \{-L, \ldots, L\}} \{s\lambda_\ell\} - p_s 4M^2 \ .$$

Applying Eq. (8) to $H(\tilde{\lambda}_{-L}^k, \ldots, \tilde{\lambda}_L^k)$ and using the fact that by construction of $\tilde{\lambda}_\ell^k$ we have $\min_\ell \left\{ \tilde{\lambda}_\ell^k \right\} = 0$ for all $k \in \mathbb{N}$ we can derive

$$\varepsilon \geq \max_\ell \left\{ \tilde{\lambda}_\ell^k \right\} - \min_\ell \left\{ \tilde{\lambda}_\ell^k \right\} - 4M^2 = \max_\ell \left\{ \tilde{\lambda}_\ell^k \right\} - 4M^2 \ .$$

We have shown that for all $k \geq K$ it holds that

$$\max_\ell \left\{ \tilde{\lambda}_\ell^k \right\} \leq 4M^2 + \varepsilon, \quad \min_\ell \left\{ \tilde{\lambda}_\ell^k \right\} = 0 \ . \tag{9}$$

Thus, for all $k \geq K$ the minimizing sequence $\tilde{\lambda}_{\ell}^k$ is bounded. Extracting convergent subsequence from $\tilde{\lambda}_{\ell}^k$, which by the abuse of notation we also denote by $\tilde{\lambda}_{\ell}^k$, and using the fact that $H : \mathbb{R}^{2L+1} \rightarrow \mathbb{R}$ is continuous we conclude that

$$\inf_{\lambda_{-L},\ldots,\lambda_L} H(\lambda_{-L},\ldots,\lambda_L) \overset{(A)}{=} \lim_{k\rightarrow\infty} H(\tilde{\lambda}_{-L}^k,\ldots,\tilde{\lambda}_L^k) \overset{(B)}{=} H(\lambda_{-L}^*,\ldots,\lambda_L^*) \; , \tag{10}$$

where $\lambda^* = (\lambda_{-L}^*,\ldots,\lambda_L^*)^\top$ is the limit of the sequence $\tilde{\lambda}_{\ell}^k$. In the above equalities $(A)$ is due to the definition of a minimizing sequence, while $(B)$ is thanks to the continuity of $H : \mathbb{R}^{2L+1} \rightarrow \mathbb{R}$. This implies that $\lambda_{-L}^*,\ldots,\lambda_L^*$ is a global minimizer. Lastly, taking the limit in Eq. (9) we conclude that for all $\varepsilon > 0$

$$\max_{\ell} \{\lambda_{\ell}^*\} \leq 4M^2 + \varepsilon, \quad \min_{\ell} \{\lambda_{\ell}^*\} = 0 \; ,$$

the proof is concluded by the fact that $\varepsilon$ is arbitrary. $\qquad\square$

# B   Preparation for fairness rates

Before establishing the main theoretical results of this work, let us introduce some notation, which compacts the proofs. We strongly suggest the reader to be familiar with this notation as it will greatly simplify the reading flow.

For all $x \in \mathbb{R}^d, s \in \mathcal{S}$ and $\ell \in \{-L,\ldots,L\}$ and all $\lambda \in \mathbb{R}$ we define

$$\hat{h}_{\ell}^s(x,\lambda) := s\lambda - \hat{p}_s(\hat{\eta}(x,s) - \ell M/L)^2 \; .$$

Therefore, using this notation, the proposed procedure $\hat{g}_L$ defined in Eq. (4) can be written as

$$\hat{g}_L(x,s) = \min\left\{ \underset{\ell \in \{-L,\ldots,L\}}{\arg\min} \left\{ -\hat{h}_{\ell}^s(x,\hat{\lambda}_{\ell}) \right\} \right\} \times \frac{M}{L} \; , \tag{11}$$

where $\hat{\lambda}_{-L},\ldots,\hat{\lambda}_L$ is a solutions of Eq. (5) rewritten as

$$\min_{\lambda_{-L},\ldots,\lambda_L} \sum_{s \in \mathcal{S}} \hat{\mathbb{E}}_{X|S=s} \left[ \max_{\ell \in \{-L,\ldots,L\}} \left\{ \hat{h}_{\ell}^s(X,\lambda_{\ell}) \right\} \right] \; . \tag{12}$$

This notation is only going to be used in the section where we derive the fairness guarantees.

In this part we also would like to introduce several standard results from empirical process theory and establish some generic properties of the minimization algorithm for $\hat{\lambda}_{-L},\ldots,\hat{\lambda}_L$ under the continuity Assumption 3.1.

**Reminder on VC theory.**

Here we remind some standard definitions of VC theory [54, 42] and already classical results from the empirical process theory on VC classes [55, 35].

**Definition B.1** (Projection). *Consider a set system $(\mathcal{X},\mathcal{R})$ with element set $\mathcal{X}$ and a set of subsets $\mathcal{R}$. Let $\mathcal{Y} \subset \mathcal{X}$ we define the projection of $\mathcal{R}$ onto $\mathcal{Y}$ as*

$$\mathcal{R}|_{\mathcal{Y}} := \{\mathcal{Y} \cap R \, : \, R \in \mathcal{R}\} \; .$$

**Definition B.2** (Shattering). *Let $(\mathcal{X},\mathcal{R})$ be a set system with element set $\mathcal{X}$ and a set of subsets $\mathcal{R}$. Let $\mathcal{Y} \subset \mathcal{X}$, we say that $\mathcal{R}$ shatters $\mathcal{Y}$ if*

$$|\mathcal{R}|_{\mathcal{Y}}| = 2^{|\mathcal{Y}|} \; ,$$

*where $|\cdot|$ stands for the cardinality when we consider sets.*

**Definition B.3** (VC-dimension). *Let $(\mathcal{X},\mathcal{R})$ be a set system. The VC-dimension of $\mathcal{R}$, denoted by $\mathrm{VC}(\mathcal{R})$ is the size of the largest subset of $\mathcal{X}$ which is shattered by $\mathcal{R}$.*

**Definition B.4** (k-Unions of ranges). *Let $(\mathcal{X},\mathcal{R})$ be a set system, for any integer $k \geq 2$, define the k-fold union of $\mathcal{R}$ as the set system induced on $\mathcal{X}$ by the ranges*

$$\mathcal{R}^{k\cup} := \{R_1 \cup \ldots \cup R_k \, : \, R_1,\ldots,R_k \in \mathcal{R}\} \; .$$

Notice that the $k$-fold union of a range set $\mathcal{R}$ are nested, that is

$$\mathcal{R} \subset \mathcal{R}^{2\cup} \subset \ldots \subset \mathcal{R}^{k\cup} \ ,$$

in particular, for all $K > 0$ it holds that

$$\bigcup_{k=1}^{K} \mathcal{R}^{k\cup} = \mathcal{R}^{K\cup} \ .$$

The next very simple result gives a bound on the VC-dimension of $k$-union of a particular range set. General treatment of this type of questions can be found in [8, 25].

**Lemma B.5.** *Let $k \geq 2$ be a positive integer and $f : \mathbb{R}^d \to \mathbb{R}$ be a fixed function. Consider the following set system $(\mathbb{R}^d, \mathcal{R}^{k\cup})$, where $\mathcal{R}$ is defined as*

$$\mathcal{R} = \left\{ R_{w_-, w_+} \ : \ w_-, w_+ \in \mathbb{R} \right\} \ ,$$

*with $R_{w_-, w_+} = \left\{ x \in \mathbb{R}^d \ : \ w_- > f(x) > w_+ \right\}$. Then,*

$$\mathrm{VC}\left(\mathcal{R}^{k\cup}\right) \leq 2k \ .$$

*Proof.* Let $\mathcal{Y} = \{x_1, \ldots, x_{2k}, x_{2k+1}\}$ be any subset of $\mathbb{R}^d$ of cardinality $2k + 1$. *W.l.o.g* suppose that

$$f(x_1) \geq \ldots \geq f(x_{2k}) \geq f(x_{2k+1}) \ .$$

Clearly, the set $\{x_1, x_3, x_5, \ldots, x_{2k+1}\}$ cannot be obtained by intersecting $\mathcal{Y}$ with any $R \in \mathcal{R}^{k\cup}$, therefore $\mathcal{Y}$ is not shattered by $\mathcal{R}^{k\cup}$. $\square$

The next result is classical and is typically derived using the entropy integral [23] combined with the Haussler's lemma [30].

**Theorem B.6** ([55]). *Let $X, X_1, \ldots, X_n$ be i.i.d. random variables distributed according to $\mathbb{P}$ on $\mathbb{R}^d$ and $(\mathbb{R}^d, \mathcal{R})$ be a range system of VC-dimension $V$, then there exists a universal constant $C > 0$ such that*

$$\mathbb{E} \sup_{R \in \mathcal{R}} \left| (\mathbb{P} - \hat{\mathbb{P}}) \mathbf{1}_{\{X \in R\}} \right| \leq C \sqrt{\frac{V}{n}} \ ,$$

*where the expectation is taken w.r.t. the joint distribution of $X_1, \ldots, X_n$, and $\hat{\mathbb{P}}$ is the empirical distribution on $X_1, \ldots, X_n$.*

**Some properties of the minimization problem in Equation** (5).

Let $P$ be a finite set of points from $\mathbb{R}^d$, we denote by $\mathrm{Co}(P)$ its convex hull. The next lemma gives the first order optimality condition for the minimization problem in Equation (5).

**Lemma B.7.** *Any solution $\hat{\lambda}_{-L}, \ldots, \hat{\lambda}_L$ of the minimization problem in Equation (5) satisfies for each $\ell \in \{-L, \ldots, L\}$*

$$0 \in \sum_{s \in \mathcal{S}} s \hat{\mathbb{P}}_{X|S=s} \left( \forall j \neq \ell \ \hat{h}_\ell^s(X, \hat{\lambda}_\ell) > \hat{h}_j^s(X, \hat{\lambda}_j) \right)$$

$$+ \sum_{s \in \mathcal{S}} \mathrm{Co}\left(\{0, s\}\right) \hat{\mathbb{P}}_{X|S=s} \left( \forall j \neq \ell \ \hat{h}_\ell^s(X, \hat{\lambda}_\ell) \geq \hat{h}_j^s(X, \hat{\lambda}_j), \ \exists j \neq \ell \ \hat{h}_\ell^{-1}(X, \hat{\lambda}_\ell) = \hat{h}_j^s(X, \hat{\lambda}_j) \right) \ .$$

*Proof.* Fix an arbitrary $\ell \in \{-L, \ldots, L\}$. For all $j \in \{-L, \ldots, L\}, x \in \mathbb{R}^d$, and $s \in \mathcal{S}$ it holds that

$$\partial_{\lambda_\ell} \hat{h}_j^s(x, \lambda_j) = s\delta_{lj} \ ,$$

where $\delta_{lj}$ is the Kronecker symbol. Thus, the subdifferential of $\max_{j \in \{-L, \ldots, L\}} \left\{ \hat{h}_j^s(X, \lambda_j) \right\}$ w.r.t. $\lambda_\ell$ is given by

$$\partial_{\lambda_\ell} \left( \max_{j \in \{-L, \ldots, L\}} \left\{ \hat{h}_j^s(x, \lambda_j) \right\} \right) = s \mathbf{1}_{\left\{ \forall j \neq \ell \ \hat{h}_\ell^s(x, \lambda_\ell) > \hat{h}_j^s(x, \lambda_j) \right\}}$$

$$+ \mathrm{Co}\left(\{0, s\}\right) \mathbf{1}_{\left\{ \forall j \neq \ell \ \hat{h}_\ell^s(x, \hat{\lambda}_\ell) \geq \hat{h}_j^s(x, \hat{\lambda}_j), \ \exists j \neq \ell \ \hat{h}_\ell^s(x, \hat{\lambda}_\ell) = \hat{h}_j^s(x, \hat{\lambda}_j) \right\}} \ .$$

We conclude the proof using the linearity of the empirical expectation and applying the first order optimality condition for convex non-differentiable problems. $\square$

The next Lemma is used to bound the second term on the right hand side of Lemma B.7. The proof of this result heavily relies on Assumption 3.1.

**Lemma B.8.** *Let Assumption 3.1 be satisfies, then for all $\ell \in \{-L, \ldots, L\}$, all $\lambda_{-L}, \ldots, \lambda_L \in \mathbb{R}$, and all $s \in \mathcal{S}$ it holds that*

$$\hat{\mathbb{P}}_{X|S=s} \left( \exists j \neq \ell \ \hat{h}_\ell^s(X, \lambda_\ell) = \hat{h}_j^s(X, \lambda_j) \right) \leq \frac{2L}{N_s} \ ,$$

*almost surely.*

*Proof.* We provide the proof for $s = 1$ and the proof for $s = -1$ follows the same arguments line by line. Fix an arbitrary $\ell \in \{-L, \ldots, L\}$ and $\lambda_{-L}, \ldots, \lambda_L \in \mathbb{R}$. If $2L \geq N_1$, then the bound is trivial, thus *w.l.o.g.*, we can assume that $2L + 1 \leq N_1$. Recall that by defintion we have

$$\hat{\mathbb{P}}_{X|S=1} \left( \exists j \neq \ell \ \hat{h}_\ell^1(X, \lambda_\ell) = \hat{h}_j^1(X, \lambda_j) \right) = \frac{1}{N_1} \sum_{X \in \mathcal{D}'_{N_1}} \mathbf{1}_{\left\{ \exists j \neq \ell \ \hat{h}_\ell^1(X, \lambda_\ell) = \hat{h}_j^1(X, \lambda_j) \right\}} \ .$$

The proof goes by contradiction. Assume that

$$\frac{1}{N_1} \sum_{X \in \mathcal{D}'_{N_1}} \mathbf{1}_{\left\{ \exists j \neq \ell \ \hat{h}_\ell^1(X, \lambda_\ell) = \hat{h}_j^1(X, \lambda_j) \right\}} \geq \frac{2L + 1}{N_1} \ ,$$

with non-zero probability. It implies that in the sum on the left hand side there are at least $2L + 1$ terms, which are exactly equal to one, while in the set $\{-L, \ldots, L\} \setminus \{\ell\}$ there are only $2L$ elements. Applying the pingeonhole principle we can conclude that there exists $j \in \{-L, \ldots, L\} \setminus \{\ell\}$ and $X, X' \in \mathcal{D}'_{N_1}$ such that simultaneously

$$\hat{h}_\ell^1(X, \lambda_\ell) = \hat{h}_j^1(X, \lambda_j)$$
$$\hat{h}_\ell^1(X', \lambda_\ell) = \hat{h}_j^1(X', \lambda_j) \ .$$

Recall that

$$\hat{h}_j^1(x, \lambda) := \lambda - \hat{p}_s(\hat{\eta}(x, 1) - jM/L)^2 \ .$$

Thus, the above two equations become:

$$\lambda_\ell - \hat{p}_1(\hat{\eta}(X, 1) - \ell M/L)^2 = \lambda_j - \hat{p}_1(\hat{\eta}(X, 1) - jM/L)^2$$
$$\lambda_\ell - \hat{p}_1(\hat{\eta}(X', 1) - \ell M/L)^2 = \lambda_j - \hat{p}_1(\hat{\eta}(X', 1) - jM/L)^2 \ .$$

Solving the above equalities for $\hat{\eta}(X, 1)$ and $\hat{\eta}(X', 1)$ implies that

$$\hat{\eta}(X, 1) = \hat{\eta}(X', 1) \ .$$

Since $X$ and $X'$ are sampled from $\mathbb{P}_{X|S=1}$, the above arguments imply that the following bound holds

$$0 < \mathbb{P} \left( \frac{1}{N_1} \sum_{X \in \mathcal{D}'_{N_1}} \mathbf{1}_{\left\{ \exists j \neq \ell \ \hat{h}_\ell^1(X, \lambda_\ell) = \hat{h}_j^1(X, \lambda_j) \right\}} \geq \frac{2L + 1}{N_1} \right)$$
$$\leq \mathbb{P} \left( \exists X, X' \in \mathcal{D}'_{N_1} \ \hat{\eta}(X, 1) = \hat{\eta}(X', 1) \right) \ .$$

Finally, notice that thanks to the continuity assumption, the random variable $\hat{\eta}(X, 1)$ almost surely does not have any atoms *w.r.t.* the measure $\mathbb{P}_{X|S=1}$, which implies that

$$\mathbb{P} \left( \exists X, X' \in \mathcal{D}'_{N_1} \ \hat{\eta}(X, 1) = \hat{\eta}(X', 1) \right) = 0 \ ,$$

and we arrive to a contradiction. $\square$

## B.1 Rates for fairness

We are now in position to prove Theorem 3.2, one of the main theoretical results of this work. Let us recall its statement in a slightly more general form.

**Theorem B.9.** *Under Assumption 3.1, there exists a universal constant $C > 0$ such that for each Borel set $\mathcal{C} \subset \mathbb{R}$ it holds that*

$$\mathbb{E}_{(\mathcal{D}_n, \mathcal{D}'_N)} \underbrace{\left| \mathbb{P}_{X|S=1} \left( \hat{g}(X,1) \in \mathcal{C} \right) - \mathbb{P}_{X|S=-1} \left( \hat{g}(X,-1) \in \mathcal{C} \right) \right|}_{\mathcal{U}(\hat{g}, \mathcal{C})} \leq C \sum_{s \in \mathcal{S}} \left( \sqrt{\frac{|\mathcal{M}|}{p_s N}} + \frac{|\mathcal{M}| L}{p_s N} \right) \quad,$$

*where $\mathcal{M} = \frac{L}{M} \times \left( \{ -L, -\frac{(L-1)M}{L}, \ldots, \frac{(L-1)M}{L}, L \} \cap \mathcal{C} \right)$. Moreover, under the same assumptions there exists a universal constant $C'$ such that*

$$\mathbb{E}_{(\mathcal{D}_n, \mathcal{D}'_N)} \sup_{\mathcal{C} \subset \mathbb{R}} \underbrace{\left| \mathbb{P}_{X|S=1} \left( \hat{g}(X,1) \in \mathcal{C} \right) - \mathbb{P}_{X|S=-1} \left( \hat{g}(X,-1) \in \mathcal{C} \right) \right|}_{\mathcal{U}(\hat{g}, \mathcal{C})} \leq C' \sum_{s \in \mathcal{S}} \left( \sqrt{\frac{L}{p_s N}} + \frac{L^2}{p_s N} \right) \quad.$$

*Proof of Theorem 3.2.* Fix some Borel subset $\mathcal{C} \subset \mathbb{R}$. First notice that thanks to the continuity assumption 3.1 it holds for all $s \in \mathcal{S}$ and all $\ell \in \{-L, \ldots, L\}$ that

$$\mathbb{P}_{X|S=s} \left( \hat{g}(X,s) = \frac{\ell M}{L} \right) = \mathbb{P}_{X|S=s} \left( \forall j \neq \ell \; \hat{h}_\ell^s(X, \hat{\lambda}_\ell) > \hat{h}_j^s(X, \hat{\lambda}_j) \right) \quad,$$

almost surely. Denote by $\mathcal{M} = \frac{L}{M} \times \left( \{ -M, -\frac{(L-1)M}{L}, \ldots, \frac{(L-1)M}{L}, M \} \cap \mathcal{C} \right)$, the scaling of those points in the grid $\mathcal{Q}_L = \{ -M, -\frac{(L-1)M}{L}, \ldots, \frac{(L-1)M}{L}, M \}$ which end up in $\mathcal{C}$, thus we can write

$$\mathbb{P}_{X|S=s} \left( \hat{g}(X,s) \in \mathcal{C} \right) = \mathbb{P}_{X|S=s} \left( \bigcup_{\ell \in \mathcal{M}} \left\{ \hat{g}(X,s) = \frac{\ell M}{L} \right\} \right)$$

$$= \mathbb{P}_{X|S=s} \left( \bigcup_{\ell \in \mathcal{M}} \left\{ \forall j \neq \ell \; \hat{h}_\ell^s(X, \hat{\lambda}_\ell) > \hat{h}_j^s(X, \hat{\lambda}_j) \right\} \right) \quad.$$

Therefore, the unfairness $\mathcal{U}(\hat{g}, \mathcal{C})$ can be written as

$$\mathcal{U}(\hat{g}, \mathcal{C}) = \left| \sum_{s \in \mathcal{S}} s \mathbb{P}_{X|S=s} \left( \bigcup_{\ell \in \mathcal{M}} \left\{ \forall j \neq \ell \; \hat{h}_\ell^s(X, \hat{\lambda}_\ell) > \hat{h}_j^s(X, \hat{\lambda}_j) \right\} \right) \right| \quad,$$

and first of all we are interested in a bound on $\mathcal{U}(\hat{g}, \mathcal{C})$ which holds almost surely.

Using the first order optimality condition for the problem in Eq. (5), derived in Lemma B.7, we can conclude that for each $\ell \in \{-L, \ldots, L\}$ there exists $\rho_1^\ell \in [0,1]$ and $\rho_{-1}^\ell \in [-1,0]$ such that

$$0 = \sum_{s \in \mathcal{S}} s \hat{\mathbb{P}}_{X|S=s} \left( \forall j \neq \ell \; \hat{h}_\ell^s(X, \hat{\lambda}_\ell) > \hat{h}_j^s(X, \hat{\lambda}_j) \right)$$

$$+ \sum_{s \in \mathcal{S}} \rho_s^\ell \hat{\mathbb{P}}_{X|S=s} \left( \forall j \neq \ell \; \hat{h}_\ell^s(X, \hat{\lambda}_\ell) \geq \hat{h}_j^s(X, \hat{\lambda}_j), \; \exists j \neq \ell \; \hat{h}_\ell^s(X, \hat{\lambda}_\ell) = \hat{h}_j^s(X, \hat{\lambda}_j) \right) \quad.$$

Note that for each $\ell \in \{-L, \ldots, L\}$ the events $\{ \forall j \neq \ell \; \hat{h}_\ell^s(X, \hat{\lambda}_\ell) > \hat{h}_j^s(X, \hat{\lambda}_j) \}$ are disjoint. Therefore, summing the above equality over $\ell \in \mathcal{M}$ we conclude that

$$0 = \sum_{s \in \mathcal{S}} s \hat{\mathbb{P}}_{X|S=s} \left( \bigcup_{\ell \in \mathcal{M}} \left\{ \forall j \neq \ell \; \hat{h}_\ell^s(X, \hat{\lambda}_\ell) > \hat{h}_j^s(X, \hat{\lambda}_j) \right\} \right)$$

$$+ \sum_{\ell \in \mathcal{M}} \sum_{s \in \mathcal{S}} \rho_s^\ell \hat{\mathbb{P}}_{X|S=s} \left( \forall j \neq \ell \; \hat{h}_\ell^s(X, \hat{\lambda}_\ell) \geq \hat{h}_j^s(X, \hat{\lambda}_j), \; \exists j \neq \ell \; \hat{h}_\ell^s(X, \hat{\lambda}_\ell) = \hat{h}_j^s(X, \hat{\lambda}_j) \right) \quad.$$

The later implies that $\mathcal{U}(\hat{g}, \mathcal{C})$ can be bounded as

$$
\mathcal{U}(\hat{g}, \mathcal{C}) \leq \sum_{s \in \mathcal{S}} \left| \left( \mathbb{P}_{X|S=s} - \hat{\mathbb{P}}_{X|S=s} \right) \mathbf{1}_{\left\{ \bigcup_{\ell \in \mathcal{M}} \left\{ \forall j \neq \ell \ \hat{h}_{\ell}^{s}(X, \hat{\lambda}_{\ell}) > \hat{h}_{j}^{s}(X, \hat{\lambda}_{j}) \right\} \right\}} \right|
$$
$$
+ \sum_{\ell \in \mathcal{M}} \sum_{s \in \mathcal{S}} \hat{\mathbb{P}}_{X|S=s} \left( \forall j \neq \ell \ \hat{h}_{\ell}^{s}(X, \hat{\lambda}_{\ell}) \geq \hat{h}_{j}^{s}(X, \hat{\lambda}_{j}), \ \exists j \neq \ell \ \hat{h}_{\ell}^{s}(X, \hat{\lambda}_{\ell}) = \hat{h}_{j}^{s}(X, \hat{\lambda}_{j}) \right)
$$
$$
\leq \sum_{s \in \mathcal{S}} \left| \left( \mathbb{P}_{X|S=s} - \hat{\mathbb{P}}_{X|S=s} \right) \mathbf{1}_{\left\{ \bigcup_{\ell \in \mathcal{M}} \left\{ \forall j \neq \ell \ \hat{h}_{\ell}^{s}(X, \hat{\lambda}_{\ell}) > \hat{h}_{j}^{s}(X, \hat{\lambda}_{j}) \right\} \right\}} \right|
$$
$$
+ \sum_{\ell \in \mathcal{M}} \sum_{s \in \mathcal{S}} \hat{\mathbb{P}}_{X|S=s} \left( \exists j \neq \ell \ \hat{h}_{\ell}^{s}(X, \hat{\lambda}_{\ell}) = \hat{h}_{j}^{s}(X, \hat{\lambda}_{j}) \right) \ .
$$

Lemma B.8 allows to control the second term on the r.h.s. of the above inequality. Thus, applying the result of Lemma B.8 and taking supremum over all $\lambda_{-L}, \ldots, \lambda_{L}$ in the first term on the r.h.s. we arrive at

$$
\mathcal{U}(\hat{g}, \mathcal{C}) \leq \sum_{s \in \mathcal{S}} \sup_{\lambda \in \mathbb{R}^{2L+1}} \left| \left( \mathbb{P}_{X|S=s} - \hat{\mathbb{P}}_{X|S=s} \right) \mathbf{1}_{\left\{ \bigcup_{\ell \in \mathcal{M}} \left\{ \forall j \neq \ell \ \hat{h}_{\ell}^{s}(X, \lambda_{\ell}) > \hat{h}_{j}^{s}(X, \lambda_{j}) \right\} \right\}} \right|
$$
$$
+ 2 \left| \mathcal{M} \right| L \left( \frac{1}{N_{-1}} + \frac{1}{N_{1}} \right) \ ,
$$

almost surely. Thus, to bound the expected value of $\mathcal{U}(\hat{g}, \mathcal{C})$ it remains to bound the expected deviation of the empirical process above and $\mathbb{E}_{(\mathcal{D}_n, \mathcal{D}_N')}[1/N_s]$ for all $s \in \mathcal{S}$.

We start by bounding the empirical process. As before, we focus on $s = 1$ and the proof for $s = -1$ is identical. To this end, for a fixed $\ell \in \mathcal{M}$, let us examine the event $\left\{ \forall j \neq \ell \ \hat{h}_{\ell}^{1}(X, \lambda_{\ell}) > \hat{h}_{j}^{1}(X, \lambda_{j}) \right\}$. Using the definition oh $\hat{h}_{j}^{1}$ we can write

$$
\left\{ \forall j \neq \ell \ \hat{h}_{\ell}^{1}(X, \lambda_{\ell}) > \hat{h}_{j}^{1}(X, \lambda_{j}) \right\} \Leftrightarrow \left\{ \forall j \neq \ell \ \lambda_{\ell} - \hat{p}_{1}(\hat{\eta}(X, 1) - \ell M/L)^{2} > \lambda_{j} - \hat{p}_{1}(\hat{\eta}(X, 1) - jM/L)^{2} \right\} \ .
$$

Rewriting the condition on the right hand side of the equivalence above we arrive at

$$
\left\{ \forall j \neq \ell \ \hat{h}_{\ell}^{1}(X, \lambda_{\ell}) > \hat{h}_{j}^{1}(X, \lambda_{j}) \right\} \Leftrightarrow \left\{ \forall j \neq \ell \ \frac{(\lambda_{\ell} - \lambda_{j}) L}{2M \hat{p}_{1}} - \frac{(\ell^{2} - j^{2}) M}{2L} > \hat{\eta}(X, 1)(j - \ell) \right\} \ .
$$

Denote by $\theta_{j}^{\ell} = \theta_{j}^{\ell}(\lambda_{-L}, \ldots, \lambda_{L}) := \frac{(\lambda_{\ell} - \lambda_{j}) L}{2M \hat{p}_{1}} - \frac{(\ell^{2} - j^{2}) M}{2L}$, thus we have

$$
\left\{ \forall j \neq \ell \ \hat{h}_{\ell}^{1}(X, \lambda_{\ell}) > \hat{h}_{j}^{1}(X, \lambda_{j}) \right\} \Leftrightarrow \left\{ \forall j \neq \ell \ \theta_{j}^{\ell} > \hat{\eta}(X, 1)(j - \ell) \right\}
$$
$$
\Leftrightarrow \left\{ \forall j > \ell \ \frac{\theta_{j}^{\ell}}{j - \ell} > \hat{\eta}(X, 1) \right\} \cap \left\{ \forall j < \ell \ \frac{\theta_{j}^{\ell}}{j - \ell} < \hat{\eta}(X, 1) \right\}
$$
$$
\Leftrightarrow \left\{ \min_{j > \ell} \frac{\theta_{j}^{\ell}}{j - \ell} > \hat{\eta}(X, 1) > \max_{j < \ell} \frac{\theta_{j}^{\ell}}{j - \ell} \right\} \ .
$$

Denoting by $w_{+}^{\ell} = w_{+}^{\ell}(\lambda_{-L}, \ldots, \lambda_{L}) = \min_{j > \ell} \frac{\theta_{j}^{\ell}}{j - \ell}$ and by $w_{-}^{\ell} = w_{-}^{\ell}(\lambda_{-L}, \ldots, \lambda_{L}) = \max_{j < \ell} \frac{\theta_{j}^{\ell}}{j - \ell}$ we get

$$
\left\{ \forall j \neq \ell \ \hat{h}_{\ell}^{1}(X, \lambda_{\ell}) > \hat{h}_{j}^{1}(X, \lambda_{j}) \right\} \Leftrightarrow \left\{ w_{+}^{\ell} > \hat{\eta}(X, 1) > w_{-}^{\ell} \right\} \ .
$$

Thus, we have

$$
\sup_{\lambda} \left| \left( \mathbb{P}_{X|S=1} - \hat{\mathbb{P}}_{X|S=1} \right) \mathbf{1}_{\left\{ \bigcup_{\ell \in \mathcal{M}} \left\{ \forall j \neq \ell \ \hat{h}_{\ell}^{1}(X, \lambda_{\ell}) > \hat{h}_{j}^{1}(X, \lambda_{j}) \right\} \right\}} \right|
$$
$$
\leq \sup_{(w_{+}^{-L}, w_{-}^{-L}), \ldots, (w_{+}^{L}, w_{-}^{L}) \in \mathbb{R}^{2}} \left| \left( \mathbb{P}_{X|S=1} - \hat{\mathbb{P}}_{X|S=1} \right) \mathbf{1}_{\left\{ \bigcup_{\ell \in \mathcal{M}} \left\{ w_{+}^{\ell} > \hat{\eta}(X, 1) > w_{-}^{\ell} \right\} \right\}} \right| \ .
$$

This implies that for all $\mathcal{C}$ it holds that

$$\mathcal{U}(\hat{g}, \mathcal{C}) \leq \sum_{s \in \mathcal{S}} \left( \sup_{(w_+^{-L}, w_-^{-L}), \ldots, (w_+^L, w_-^L) \in \mathbb{R}^2} \left| \left( \mathbb{P}_{X|S=1} - \hat{\mathbb{P}}_{X|S=1} \right) \mathbf{1}_{\{\bigcup_{\ell \in \mathcal{M}} \{w_+^\ell > \hat{\eta}(X,1) > w_-^\ell\}\}} \right| + 2 |\mathcal{M}| L N_s^{-1} \right) .$$

We are ready to prove the **first claim** of the result. Combining Lemma B.5 with Lemma B.6 we conclude that there exists $C > 0$ such that

$$\mathbb{E} \left[ \sup_{(w_+^{-L}, w_-^{-L}), \ldots, (w_+^L, w_-^L) \in \mathbb{R}^2} \left| \left( \mathbb{P}_{X|S=1} - \hat{\mathbb{P}}_{X|S=1} \right) \mathbf{1}_{\{\bigcup_{\ell \in \mathcal{M}} \{w_+^\ell > \hat{\eta}(X,1) > w_-^\ell\}\}} \right| \, \Big| \, \mathcal{D}_N^S, \mathcal{D}_n \right] \leq C \sqrt{\frac{2 |\mathcal{M}|}{N_1}} .$$

Finally, repeating the same argument for $s = -1$ we obtain for some universal $C > 0$

$$\mathbb{E}_{(\mathcal{D}_n, \mathcal{D}_N')}[\mathcal{U}(\hat{g}, \mathcal{C})] \leq C \mathbb{E} \left( \sqrt{\frac{2 |\mathcal{M}|}{N_{-1}}} + \sqrt{\frac{2 |\mathcal{M}|}{N_1}} \right) + 2 \mathbb{E} \left( \frac{|\mathcal{M}| L}{N_{-1}} + \frac{|\mathcal{M}| L}{N_1} \right) .$$

Note that $N_{-1}$ and $N_1$ are binomial random variables with parameters $(p_{-1}, N)$ and $(p_1, N)$ respectively. Applying the bound on the moment of binomials random variables we conclude that for some universal $C > 0$ it holds that

$$\mathbb{E}_{(\mathcal{D}_n, \mathcal{D}_N')}[\mathcal{U}(\hat{g}, \mathcal{C})] \leq C \sum_{s \in \mathcal{S}} \left( \sqrt{\frac{|\mathcal{M}|}{p_s N}} + \frac{|\mathcal{M}| L}{p_s N} \right) .$$

In order to prove the **second claim** of the result, we first notice that following the same argument we can write

$$\sup_{\mathcal{C} \subset \mathbb{R}} \mathcal{U}(\hat{g}, \mathcal{C}) \leq \sum_{s \in \mathcal{S}} \left( \sup_{R \in \mathcal{R}_s} \left| \left( \mathbb{P}_{X|S=s} - \hat{\mathbb{P}}_{X|S=s} \right) \mathbf{1}_{\{X \in R\}} \right| + \frac{4 L^2}{N_s} \right) ,$$

almost surely. Here, for all $s \in \mathcal{S}$ the range set $\mathcal{R}_s$ is defined as

$$\mathcal{R}_s = \bigcup_{\ell=1}^{2L+1} \mathcal{R}_{\hat{\eta}, s}^{\ell \cup} ,$$

where $\mathcal{R}_{\hat{\eta}, s} = \left\{ R_{a,b}^s \, : \, a, b \in \mathbb{R} \right\}$ and $R_{a,b}^s = \left\{ x \in \mathbb{R}^d \, : \, a > \hat{\eta}(x, s) > b \right\}$. In words, the ranges of $\mathcal{R}_s$ are induced by $2L + 1$-fold union of level sets of $\hat{\eta}(\cdot, s)$, with $\hat{\eta}(\cdot, s)$ being fixed conditionally on the labeled dataset. Note that again thanks to Lemma B.5 and the inclusion of $k$-fold unions it holds that

$$\mathrm{VC}(\mathcal{R}_s) \leq 2L + 1 ,$$

for all $s \in \mathcal{S}$. We conclude similarly to the previous case applying Lemma B.6, which formally replaces $|\mathcal{M}|$ by $2L + 1$. $\qquad\square$

## C  Preparation for risk rates

As in the previous part, we first present some preparation results which allow to establish the consistency of the proposed procedure in terms of the risk measure. We suggest the reader to understand the statements the following lemmas first and immediately proceed to the proof of the risk consistency result. After the proof of the main result, the interested reader could proceed to the proofs of the lemmas of this section.

The next tautology is used to simplify the presentation.

**Lemma C.1.** *For any g it holds that*

$$\mathcal{R}(g) = \mathbb{E}[Y^2] - \mathbb{E}[\eta^2(X, S)] + \sum_{s \in \mathcal{S}} p_s \mathbb{E}_{X|S=s} (\eta(X, s) - g(X, s))^2 .$$

Let $r(\cdot)$ be defined as

$$r(g) := \sum_{s \in \mathcal{S}} p_s \mathbb{E}_{X|S=s} (\eta(X,s) - g(X,s))^2 = \mathbb{E}_{(X,S)} (\eta(X,S) - g(X,S))^2 \ .$$

Notice that for any $g, g'$ it holds that

$$\mathcal{R}(g) - \mathcal{R}(g') = r(g) - r(g') \ ,$$

therefore, from now on we focus on $r(\hat{g}) - r(g^*)$ instead of $\mathcal{R}(\hat{g}) - \mathcal{R}(g^*)$.

The next result provides an alternative expression for the risk of the oracle $g^*$.

**Lemma C.2.** *Let the continuity Assumption 2.5 be satisfied, then*

$$r(g^*) = \max_{\lambda \in \mathbb{R}^{2L+1}} \sum_{s \in \mathcal{S}} \mathbb{E}_{X|S=s} \min_{\ell \in \{-L,\dots,L\}} \left\{ -s\lambda_\ell + p_s \left( \eta(X,s) - \frac{\ell M}{L} \right)^2 \right\} \ .$$

We also need a suitable upper bound on the risk of the proposed procedure $\hat{g}$, which is derived very similarly to Lemma C.2.

**Lemma C.3.** *The proposed estimator $\hat{g}$ satisfies almost surely*

$$r(\hat{g}) \leq \sum_{s \in \mathcal{S}} \mathbb{E}_{X|S=s} \min_{\ell \in \{-L,\dots,L\}} \left\{ -s\hat{\lambda}_\ell + \hat{p}_s \left( \hat{\eta}(X,s) - \frac{\ell M}{L} \right)^2 \right\}$$

$$+ \sum_{\ell=-L}^{L} \hat{\lambda}_\ell \sum_{s \in \mathcal{S}} s \mathbb{P}_{X|S=s} \left( \hat{g}(X,s) = \frac{\ell M}{L} \right) + 4M \|\eta - \hat{\eta}\|_1 + 4M^2 \sum_{s \in \mathcal{S}} |p_s - \hat{p}_s| \ ,$$

*where $\|\eta - \hat{\eta}\|_1 = \mathbb{E}_{(X,S)} |\eta(X,S) - \hat{\eta}(X,S)|$.*

There are four terms in the expression for $r(\hat{g})$: the first one is the risk of $\hat{g}$ if the practitioner had access to the marginal distribution of $(X,S)$; the second term described the violation of the fairness constraints; the third is coming from the fact that we use $\hat{\eta}$ in place of $\eta$; the last term appears due to estimation of the marginal distribution of $S$. Equipped with the two above results we deduce the following corollary on the excess risk of the proposed procedure.

**Corollary C.4.** *Under Assumption 2.5 the proposed estimator $\hat{g}$ satisfies almost surely*

$$r(\hat{g}) - r(g^*) \leq 8M \|\eta - \hat{\eta}\|_1 + 8M^2 \sum_{s \in \mathcal{S}} |p_s - \hat{p}_s| + \sum_{\ell=-L}^{L} \hat{\lambda}_\ell \sum_{s \in \mathcal{S}} s \mathbb{P}_{X|S=s} \left( \hat{g}(X,s) = \frac{\ell M}{L} \right) \ .$$

*Proof.* Let us introduce some short-hand notation to save space

$$\alpha = \sum_{s \in \mathcal{S}} \mathbb{E}_{X|S=s} \min_{\ell \in \{-L,\dots,L\}} \left\{ -s\hat{\lambda}_\ell + \hat{p}_s \left( \hat{\eta}(X,s) - \frac{\ell M}{L} \right)^2 \right\}$$

$$\beta = \max_{\lambda} \sum_{s \in \mathcal{S}} \mathbb{E}_{X|S=s} \min_{\ell \in \{-L,\dots,L\}} \left\{ -s\lambda_\ell + p_s \left( \eta(X,s) - \frac{\ell M}{L} \right)^2 \right\} \ .$$

Using the above we can write

$$\alpha - \beta \leq \sum_{s \in \mathcal{S}} \mathbb{E}_{X|S=s} \min_{\ell \in \{-L,\dots,L\}} \left\{ -s\hat{\lambda}_\ell + \hat{p}_s \left( \hat{\eta}(X,s) - \frac{\ell M}{L} \right)^2 \right\}$$

$$- \sum_{s \in \mathcal{S}} \mathbb{E}_{X|S=s} \min_{\ell \in \{-L,\dots,L\}} \left\{ -s\hat{\lambda}_\ell + p_s \left( \eta(X,s) - \frac{\ell M}{L} \right)^2 \right\}$$

$$\leq \sum_{s \in \mathcal{S}} \mathbb{E}_{X|S=s} \min_{\ell \in \{-L,\dots,L\}} \left\{ -s\hat{\lambda}_\ell + p_s \left( \hat{\eta}(X,s) - \frac{\ell M}{L} \right)^2 \right\} + 4M^2 \sum_{s \in \mathcal{S}} |p_s - \hat{p}_s|$$

$$- \sum_{s \in \mathcal{S}} \mathbb{E}_{X|S=s} \min_{\ell \in \{-L,\dots,L\}} \left\{ -s\hat{\lambda}_\ell + p_s \left( \eta(X,s) - \frac{\ell M}{L} \right)^2 \right\}$$

$$\leq \sum_{s \in \mathcal{S}} p_s \mathbb{E}_{X|S=s} \max_\ell \left| \left( \hat{\eta}(X,s) - \frac{\ell M}{L} \right)^2 - \left( \eta(X,s) - \frac{\ell M}{L} \right)^2 \right| + 4M^2 \sum_{s \in \mathcal{S}} |p_s - \hat{p}_s|$$

$$\leq 4M \sum_{s \in \mathcal{S}} p_s \mathbb{E}_{X|S=s} |\hat{\eta}(X,s) - \eta(X,s)| + 4M^2 \sum_{s \in \mathcal{S}} |p_s - \hat{p}_s|$$

$$= 4M \|\eta - \hat{\eta}\|_1 + 4M^2 \sum_{s \in \mathcal{S}} |p_s - \hat{p}_s| \quad .$$

Finally, combining Lemma C.2 with Lemma C.3 implies the statement of the corollary. $\qquad\square$

## C.1 Rates for the excess risk

We are ready to present the proof of the rates of convergence of the excess risk of the proposed procedure stated in Theorem 3.3. Recall that Lemma 2.4 gives a way to control $\mathcal{R}(g_L^*) - \mathcal{R}(f^*)$. Thus, to control the excess risk of $g_L^*$ it only remains to bound $\mathcal{R}(\hat{g}_L) - \mathcal{R}(g_L^*)$. From now on we again omit the index $L$. We also recall[6] the statement of Theorem 3.3

**Theorem C.5.** *Let Assumptions 2.5 and 3.1 be satisfied, then for the proposed estimator $\hat{g}$ there exists a universal constant $C > 0$ such that*

$$\mathbb{E}_{(\mathcal{D}_n, \mathcal{D}_N')}[\mathcal{R}(\hat{g})] - \mathcal{R}(g^*) \leq 8M \mathbb{E}_{(\mathcal{D}_n, \mathcal{D}_N')} \|\eta - \hat{\eta}\|_1 + CM^2 \sum_{s \in \mathcal{S}} \left( L \sqrt{\frac{1}{p_s N}} + \frac{L^2}{p_s N} \right) \quad .$$

*Proof of Theorem C.5.* As already discussed we have

$$\mathbb{E}_{(\mathcal{D}_n, \mathcal{D}_N')}[\mathcal{R}(\hat{g})] - \mathcal{R}(g^*) = \mathbb{E}_{(\mathcal{D}_n, \mathcal{D}_N')}[r(\hat{g})] - r(g^*) \quad . \tag{13}$$

Thanks to Corollary C.4 we have

$$\mathbb{E}_{(\mathcal{D}_n, \mathcal{D}_N')}[r(\hat{g})] - r(g^*) \leq \sum_{\ell=-L}^{L} \mathbb{E}_{(\mathcal{D}_n, \mathcal{D}_N')} \left[ \hat{\lambda}_\ell \sum_{s \in \mathcal{S}} s \mathbb{P}_{X|S=s} \left( \hat{g}(X,s) = \frac{\ell M}{L} \right) \right]$$

$$+ 8M \mathbb{E}_{(\mathcal{D}_n, \mathcal{D}_N')} \|\eta - \hat{\eta}\|_1 + 8M^2 \sum_{s \in \mathcal{S}} \mathbb{E}_{(\mathcal{D}_n, \mathcal{D}_N')} |\hat{p}_s - p_s| \quad .$$

Let us bound the first term on the right hand side of the above inequality. Thanks to Proposition A.2 we know that for all $\ell \in \{-L, \dots, L\}$ it holds that $|\hat{\lambda}_\ell| \leq 4M^2$. Note that Proposition A.2 is proven for $\lambda^*$, yet an identical proof yields the same conclusion on $\hat{\lambda}$. Using this we can write introducing the notation

$$(*) = \sum_{\ell=-L}^{L} \mathbb{E}_{(\mathcal{D}_n, \mathcal{D}_N')} \left[ \hat{\lambda}_\ell \sum_{s \in \mathcal{S}} s \mathbb{P}_{X|S=s} \left( \hat{g}(X,s) = \frac{\ell M}{L} \right) \right] \quad ,$$

that

$$(*) \leq 4M^2 \sum_{\ell=-L}^{L} \mathbb{E}_{(\mathcal{D}_n, \mathcal{D}'_N)} \left| \sum_{s \in \mathcal{S}} s \mathbb{P}_{X|S=s} \left( \hat{g}(X, s) = \frac{\ell M}{L} \right) \right| .$$

For each $\ell \in \{-L, \dots, L\}$ we can use Theorem B.9 with $|\mathcal{M}| = 1$ which implies that for some universal constant $C > 0$ we have

$$(*) \leq CM^2 \sum_{s \in \mathcal{S}} \left( L \sqrt{\frac{1}{p_s N}} + \frac{L^2}{p_s N} \right) .$$

Finally, we can write for some universal $C > 0$ that

$$\sum_{s \in \mathcal{S}} \mathbb{E}_{(\mathcal{D}_n, \mathcal{D}'_N)} |\hat{p}_s - p_s| = 2 \mathbb{E}_{(\mathcal{D}_n, \mathcal{D}'_N)} |p_1 - \hat{p}_1| \leq C \sqrt{\frac{1}{N}} .$$

Combining all of the above we conclude. $\qquad\square$

The proof of Theorem 3.3 ends if we combine Theorem C.5 with Lemma 2.4..

## C.2 Proofs of preparation results

*Proof of Lemma C.2.* We have the following chain of equalities

$$r(g^*) = \sum_{s \in \mathcal{S}} p_s \mathbb{E}_{X|S=s} (\eta(X, s) - g^*(X, s))^2$$

$$= \sum_{\ell=-L}^{L} \sum_{s \in \mathcal{S}} p_s \mathbb{E}_{X|S=s} \left( \eta(X, s) - \frac{\ell M}{L} \right)^2 \mathbf{1}_{\{g^*(X,s)=\frac{\ell M}{L}\}}$$

$$= \sum_{\ell=-L}^{L} \sum_{s \in \mathcal{S}} \mathbb{E}_{X|S=s} \left( -s\lambda_\ell^* + p_s \left( \eta(X, s) - \frac{\ell M}{L} \right)^2 \right) \mathbf{1}_{\{g^*(X,s)=\frac{\ell M}{L}\}}$$

$$+ \sum_{\ell=-L}^{L} \lambda_\ell^* \sum_{s \in \mathcal{S}} s \mathbb{P}_{X|S=s} \left( g^*(X, s) = \frac{\ell M}{L} \right) .$$

Since $g^*$ is fair it holds that

$$\sum_{s \in \mathcal{S}} s \mathbb{P}_{X|S=s} \left( g^*(X, s) = \frac{\ell M}{L} \right) = 0 ,$$

for all $\ell \in \{-L, \dots, L\}$. Thus we have

$$r(g^*) = \sum_{\ell=-L}^{L} \sum_{s \in \mathcal{S}} \mathbb{E}_{X|S=s} \left( -s\lambda_\ell^* + p_s \left( \eta(X, s) - \frac{\ell M}{L} \right)^2 \right) \mathbf{1}_{\{g^*(X,s)=\frac{\ell M}{L}\}} .$$

Recall that for every $(x, s) \in \mathbb{R}^d \times \mathcal{S}$ the oracle $g^*$ is defined as

$$g^*(x, s) = \arg\min_\ell \left\{ -s\lambda_\ell^* + p_s \left( \eta(x, s) - \frac{\ell M}{L} \right)^2 \right\} \times \frac{M}{L} ,$$

thus for $r(g^*)$ we can write

$$r(g^*) = \sum_{s \in \mathcal{S}} \mathbb{E}_{X|S=s} \min_{\ell \in \{-L, \dots, L\}} \left\{ -s\lambda_\ell^* + p_s \left( \eta(X, s) - \frac{\ell M}{L} \right)^2 \right\} .$$

Using the definition of $\lambda_{-L}^*, \dots, \lambda_L^*$ we have

$$r(g^*) = \max_\lambda \sum_{s \in \mathcal{S}} \mathbb{E}_{X|S=s} \min_{\ell \in \{-L, \dots, L\}} \left\{ -s\lambda_\ell + p_s \left( \eta(X, s) - \frac{\ell M}{L} \right)^2 \right\} .$$

$\qquad\square$

*Proof of Lemma C.3.* Conditionally on all data we can write

$$r(\hat{g}) = \mathbb{E}(\hat{\eta}(X,S) - \hat{g}(X,S))^2 + \mathbb{E}(\eta(X,S) - \hat{g}(X,S))^2 - \mathbb{E}(\hat{\eta}(X,S) - \hat{g}(X,S))^2 .$$

Note that the boundedness of $Y \in \mathbb{R}$, implies the boundedness of $\eta(X,S)$. Thus, we have

$$\mathbb{E}(\eta(X,S) - \hat{g}(X,S))^2 - \mathbb{E}(\hat{\eta}(X,S) - \hat{g}(X,S))^2 \le 4M \|\eta - \hat{\eta}\|_1 .$$

So far we showed that the following bound holds almost surely

$$r(\hat{g}) \le \mathbb{E}(\hat{\eta}(X,S) - \hat{g}(X,S))^2 + 4M \|\eta - \hat{\eta}\|_1 .$$

Now, let us work with $\mathbb{E}(\hat{\eta}(X,S) - \hat{g}(X,S))^2$. We can write

$$\mathbb{E}(\hat{\eta}(X,S) - \hat{g}(X,S))^2 = \sum_{s \in \mathcal{S}} p_s \mathbb{E}_{X|S=s}(\hat{\eta}(X,s) - \hat{g}(X,s))^2$$

$$= \sum_{s \in \mathcal{S}} \hat{p}_s \mathbb{E}_{X|S=s}(\hat{\eta}(X,s) - \hat{g}(X,s))^2$$

$$+ \sum_{s \in \mathcal{S}} (p_s - \hat{p}_s) \mathbb{E}_{X|S=s}(\hat{\eta}(X,s) - \hat{g}(X,s))^2$$

$$\le \sum_{s \in \mathcal{S}} \hat{p}_s \mathbb{E}_{X|S=s}(\hat{\eta}(X,s) - \hat{g}(X,s))^2 + 4M^2 \sum_{s \in \mathcal{S}} |p_s - \hat{p}_s| .$$

Lastly, for the first term on the right hand side of the above inequality we can write

$$\sum_{s \in \mathcal{S}} \hat{p}_s \mathbb{E}_{X|S=s}(\hat{\eta}(X,s) - \hat{g}(X,s))^2 = \sum_{\ell=-L}^{L} \sum_{s \in \mathcal{S}} \mathbb{E}_{X|S=s} \hat{p}_s \left( \hat{\eta}(X,s) - \frac{\ell M}{L} \right)^2 \mathbf{1}_{\{\hat{g}(X,s) = \frac{\ell M}{L}\}}$$

$$= \sum_{\ell=-L}^{L} \sum_{s \in \mathcal{S}} \mathbb{E}_{X|S=s} \left( -s\hat{\lambda}_\ell + \hat{p}_s \left( \hat{\eta}(X,s) - \frac{\ell M}{L} \right)^2 \right) \mathbf{1}_{\{\hat{g}(X,s) = \frac{\ell M}{L}\}}$$

$$+ \sum_{\ell=-L}^{L} \hat{\lambda}_\ell \sum_{s \in \mathcal{S}} s \mathbb{P}_{X|S=s} \left( \hat{g}(X,s) = \frac{\ell M}{L} \right) .$$

Recall that for each $(x,s) \in \mathbb{R}^d \times \mathcal{S}$ the estimator $\hat{g}$ is defined as

$$\hat{g}(x,s) = \min \left\{ \arg\min_\ell \left\{ -s\hat{\lambda}_\ell + \hat{p}_s \left( \hat{\eta}(x,s) - \frac{\ell M}{L} \right)^2 \right\} \right\} \times \frac{M}{L} ,$$

thus we have

$$\sum_{s \in \mathcal{S}} \hat{p}_s \mathbb{E}_{X|S=s}(\hat{\eta}(X,s) - \hat{g}(X,s))^2 = \sum_{s \in \mathcal{S}} \mathbb{E}_{X|S=s} \min_{\ell \in \{-L,\dots,L\}} \left\{ -s\hat{\lambda}_\ell + \hat{p}_s \left( \hat{\eta}(X,s) - \frac{\ell M}{L} \right)^2 \right\}$$

$$+ \sum_{\ell=-L}^{L} \hat{\lambda}_\ell \sum_{s \in \mathcal{S}} s \mathbb{P}_{X|S=s} \left( \hat{g}(X,s) = \frac{\ell M}{L} \right) .$$

Combining all of the above concludes the proof. $\qquad\square$

# D  Optimization algorithm to approximate the thresholds

The whole section is devoted to the proof of Theorem 3.5. We denote by $\Delta$ the probability simplex in $\mathbb{R}^{2L+1}$. As pointed out, in the main body of the paper, we set $\hat{\lambda}_{-L}, \dots, \hat{\lambda}_L$ to be a solution of Eq. (5). Let us recall that the problem in Eq. (5) is an example of non-smooth convex optimization, and subgradient methods can be used to find a solution numerically. Yet, subgradient methods suffer from instability of the outcome and have slow rates of convergence. To alleviate this issue we leverage the structure of problem (5) and apply the idea of smoothing, developed in the context of optimization [45].

Thus, instead of building an iterative scheme for problem (5) we focus on its proxy-problem defined for all $\beta > 0$ as

$$\min_{\lambda_{-L},\dots,\lambda_L} \sum_{s \in \mathcal{S}} \hat{\mathbb{E}}_{X|S=s} \max_{w \in \Delta} \left\{ \sum_{\ell=-L}^{L} w_\ell \left( s\lambda_\ell - \hat{Z}_\ell(X,s) \right) - \beta \operatorname{KL}(w||\pi) \right\} , \qquad (\mathcal{P}_{\hat{\lambda}}^{\beta})$$

where $\pi = (1/(2L+1),\dots,1/(2L+1))^\top \in \mathbb{R}^{2L+1}$ and the KL-divergence is defined as

$$\operatorname{KL}(w||\pi) = \sum_{\ell=-L}^{L} w_\ell \log \frac{w_l}{\pi_l} . \qquad (14)$$

Denote by $G$ and $G_\beta$ the objective functions of the minimization problems in Eq. (5) and in $(\mathcal{P}_{\hat{\lambda}}^{\beta})$ respectively.

Therefore, $\hat{\lambda} = (\hat{\lambda}_{-L},\dots,\hat{\lambda}_L)^\top$ is defined as

$$\hat{\lambda} \in \arg\min_{\lambda \in \mathbb{R}^{2L+1}} G(\lambda) .$$

Also, define $\hat{\lambda}_\beta$ as

$$\hat{\lambda}_\beta \in \arg\min_{\lambda \in \mathbb{R}^{2L+1}} G_\beta(\lambda) .$$

The next result tells that $G_\beta$ is indeed an approximation of $G$ as long as $\beta$ is sufficiently small.

**Lemma D.1.** *For all $\lambda \in \mathbb{R}^{2L+1}$ it holds that*

$$G_\beta(\lambda) \leq G(\lambda) \leq G_\beta(\lambda) + 2\beta \log(2L+1) .$$

*Proof of Lemma D.1.* For any probability vector $w$ it holds that $0 \leq \sum_{\ell=-L}^{L} w_\ell \log \frac{w_l}{\pi_l} \leq \log(2L+1)$. Applying this fact concludes the proof. $\qquad \square$

We also need to derive an explicit expression for $G_\beta$.

**Lemma D.2.** *For any $\beta > 0$ it holds that*

$$G_\beta(\lambda) = \beta \sum_{s \in \mathcal{S}} \hat{\mathbb{E}}_{X|S=s} \log \left( \sum_{\ell=-L}^{L} \exp \left( \frac{1}{\beta} s\lambda_\ell - \frac{1}{\beta} \hat{Z}_\ell(X,s) \right) \right) - 2\beta \log(2L+1) .$$

*Proof of Lemma D.2.* For a fixed $s \in \mathcal{S}$ and a fixed $x \in \mathbb{R}^d$, let us first solve another problem, namely we would like to find a maximizer of

$$\max \left\{ \sum_{\ell=-L}^{L} w_\ell \left( s\lambda_\ell - \hat{Z}_\ell(x,s) - \beta \log \frac{w_\ell}{\pi_\ell} \right) : \sum_{\ell=-L}^{L} w_\ell = 1 \right\} . \qquad (15)$$

To solve this problem analytically, we construct the Lagrangian function as

$$\mathcal{L}(w,\kappa) = \sum_{\ell=-L}^{L} w_\ell \left( s\lambda_\ell - \hat{Z}_\ell(x,s) - \beta \log \frac{w_\ell}{\pi_\ell} \right) + \kappa \left( \sum_{\ell=-L}^{L} w_\ell - 1 \right) .$$

The KKT conditions read as

$$\partial_{w_\ell} \mathcal{L}(w,\kappa) = 0 ,$$

$$\sum_{\ell=-L}^{L} w_\ell = 1 ,$$

for all $\ell \in \{-L, \ldots, L\}$. Taking the partial derivatives we get

$$\partial_{w_\ell} \mathcal{L}(p, \kappa) = s\lambda_\ell - \hat{Z}_\ell(x, s) - \beta \log \frac{w_\ell}{\pi_\ell} - \beta + \kappa = 0 \ , \tag{16}$$

$$\sum_{\ell=-L}^{L} w_\ell = 1 \ . \tag{17}$$

Solving Eq. (16) for $w_l$ we obtain

$$- \beta \log \frac{w_\ell}{\pi_\ell} = -s\lambda_\ell + \hat{Z}_\ell(x, s) + \beta - \kappa \ ,$$

$$\log \frac{w_\ell}{\pi_\ell} = \frac{1}{\beta} s\lambda_\ell - \frac{1}{\beta} \hat{Z}_\ell(x, s) - 1 + \frac{1}{\beta} \kappa \ ,$$

$$w_\ell = \frac{1}{2L+1} \exp\left( \frac{1}{\beta} s\lambda_\ell - \frac{1}{\beta} \hat{Z}_\ell(x, s) \right) \exp\left( -1 + \frac{1}{\beta} \kappa \right) \ .$$

Using the relation in Eq. (17), we find the value of the dual variable $\kappa$ as

$$\exp\left( -1 + \frac{1}{\beta} \kappa \right) = \left( \frac{1}{2L+1} \sum_{\ell=-L}^{L} \exp\left( \frac{1}{\beta} s\lambda_\ell - \frac{1}{\beta} \hat{Z}_\ell(x, s) \right) \right)^{-1} \tag{18}$$

Plug-in the above into the expression for $w_\ell$ we arrive at

$$w_\ell = \frac{\exp\left( \frac{1}{\beta} s\lambda_\ell - \frac{1}{\beta} \hat{Z}_\ell(x, s) \right)}{\sum_{\ell=-L}^{L} \exp\left( \frac{1}{\beta} s\lambda_\ell - \frac{1}{\beta} \hat{Z}_\ell(x, s) \right)} \ .$$

Note that $w_\ell \in [0, 1]$ and $\sum_\ell w_\ell = 1$, therefore it is a minimizer of

$$\max_{w \in \Delta} \left\{ \sum_{\ell=-L}^{L} w_\ell \left( s\lambda_\ell - \hat{Z}_\ell(x, s) - \beta \log \frac{w_\ell}{\pi_\ell} \right) \right\} \ .$$

Plug-in the expression for $w_\ell$ into the above objective function we conclude that

$$\max_{w \in \Delta} \left\{ \sum_{\ell=-L}^{L} w_\ell \left( s\lambda_\ell - \hat{Z}_\ell(x, s) - \beta \log \frac{w_\ell}{\pi_\ell} \right) \right\} = \beta \log \left( \sum_{\ell=-L}^{L} \exp\left( \frac{1}{\beta} s\lambda_\ell - \frac{1}{\beta} \hat{Z}_\ell(x, s) \right) \right)$$
$$- \beta \log(2L+1) \ .$$

Thus the minimizer of problem $(\mathcal{P}_{\hat{\lambda}}^\beta)$ is also the solution of

$$\min_\lambda \left\{ \beta \sum_{s \in \mathcal{S}} \hat{\mathbb{E}}_{X|S=s} \log \left( \sum_{\ell=-L}^{L} \exp\left( \frac{1}{\beta} s\lambda_\ell - \frac{1}{\beta} \hat{Z}_\ell(X, s) \right) \right) - \beta \log(2L+1) \right\} \ .$$

Therefore,

$$G_\beta(\lambda) = \beta \sum_{s \in \mathcal{S}} \hat{\mathbb{E}}_{X|S=s} \log \left( \sum_{\ell=-L}^{L} \exp\left( \frac{1}{\beta} s\lambda_\ell - \frac{1}{\beta} \hat{Z}_\ell(X, s) \right) \right) - 2\beta \log(2L+1) \ . \tag{19}$$

$\square$

The function $G_\beta$ is appealing due to the fact that it is smooth and its gradient is Lipschitz.

**Lemma D.3** ([27]). *The function $G_\beta$ has a continuous gradient with Lipschitz constant $2/\beta$, that is, for all $\lambda, \lambda'$ it holds that*

$$\|\nabla G_\beta(\lambda) - \nabla G_\beta(\lambda')\|_2 \leq \frac{2}{\beta} \|\lambda - \lambda'\|_2 \ .$$

Note that small values of $\beta$ induce large Lipschitz constant and thus this function is harder to minimize.

Let us also derive the gradient of $G_\beta$ in order to apply iterative procedures.

**Lemma D.4.** *For every $\lambda \in \mathbb{R}^{2L+1}$, the following expression holds for the gradient of $G_\beta$*

$$\left(\nabla G_\beta(\lambda)\right)_\ell = \sum_{s \in \mathcal{S}} s \hat{\mathbb{E}}_{X|S=s} \frac{\exp\left(\frac{s}{\beta}\lambda_\ell - \frac{1}{\beta}\hat{Z}_\ell(X,s)\right)}{\sum_{\ell=-L}^{L} \exp\left(\frac{s}{\beta}\lambda_\ell - \frac{1}{\beta}\hat{Z}_\ell(X,s)\right)} \ ,$$

*for each $\ell \in \{-L, \ldots, L\}$.*

Let us recall the accelerated gradient descent for convex $(2/\beta)$ -smooth functions. The goal is to approximate

$$\min G_\beta(\lambda) \ .$$

The iterations of the accelerated gradient descent are given by

$$\lambda_1 = y_1 = \tau_0 = 0 \ ,$$
$$y_{t+1} = \lambda_t - \frac{\beta}{2}\nabla G_\beta(\lambda_t) \ ,$$
$$\lambda_{t+1} = (1-\gamma_t)y_{t+1} + \gamma_t y_t \ ,$$
$$\tau_t = \frac{1 + \sqrt{1 + 4\tau_{t-1}^2}}{2} \ ,$$
$$\gamma_t = \frac{1 - \tau_t}{\tau_{t+1}} \ .$$

The next result is already classical in the optimization literature, its proof can be found in [44, 5].

**Theorem D.5** ([44]). *The above iteration satisfies*

$$G_\beta(\lambda_T) - G_\beta(\hat{\lambda}_\beta) \le \frac{4\|\lambda_1 - \hat{\lambda}_\beta\|_2^2}{\beta T^2} \ .$$

Combination of Theorem D.5 with Lemma D.1 immediately yields.

**Corollary D.6.** *Let $\lambda_T$ be the output Algorithm 1, therefore*

$$G(\lambda_T) - G(\hat{\lambda}) \le \frac{4\|\hat{\lambda}_\beta\|_2^2}{\beta T^2} + 2\beta \log(2L+1) \ .$$

*Proof.* Thanks to Lemma D.1 we have

$$G(\lambda_T) \le G_\beta(\lambda_T) + 2\beta \log(2L+1) \ ,$$
$$G(\hat{\lambda}) \ge G_\beta(\hat{\lambda}) \ge G_\beta(\hat{\lambda}_\beta) \ .$$

Moreover, using Theorem D.5 we get

$$G(\lambda_T) - G(\hat{\lambda}) \le \frac{4\|\hat{\lambda}_\beta\|_2^2}{\beta T^2} + 2\beta \log(2L+1) \ .$$

$\square$

Let us understand the order of magnitude of $\|\hat{\lambda}_\beta\|_2^2$.

**Lemma D.7.** *For any positive $\beta$ it holds that*

$$\|\hat{\lambda}_\beta\|_\infty \le 4M^2 + 2\beta \log(2L+1) \ .$$

*Proof.* Notice that

$$G_\beta(0) \le G(0) \le 0 \ .$$

Moreover, for any $\lambda \in \mathbb{R}^{2L+1}$ we have

$$G_\beta(\lambda) = \beta \sum_{s \in \mathcal{S}} \hat{\mathbb{E}}_{X|S=s} \log \left( \sum_{\ell=-L}^{L} \exp \left( \frac{1}{\beta} s\lambda_\ell - \frac{1}{\beta} \hat{Z}_\ell(X,s) \right) \right) - 2\beta \log(2L+1)$$
$$\ge G(\lambda) - 2\beta \log(2L+1)$$
$$\ge \max \{\lambda_\ell\} - \min \{\lambda_\ell\} - 4M^2 - 2\beta \log(2L+1) \ .$$

And we conclude similarly to Proposition A.2. $\qquad\square$

**Corollary D.8.** *For any positive $\beta$ it holds that*

$$G(\lambda_T) - G(\hat{\lambda}) \le 128M^4 \frac{2L+1}{\beta T^2} + 128\beta \log^2(2L+1) \ .$$

*Proof.* Recall that for any $\lambda \in \mathbb{R}^{2L+1}$ it holds

$$\|\lambda\|_2^2 \le \|\lambda\|_\infty^2 (2L+1) \ .$$

Therefore thanks to Lemma D.7, for $\hat{\lambda}_\beta$ we have

$$\left\| \hat{\lambda}_\beta \right\|_2^2 \le (2L+1) \left( 4M^2 + 2\beta \log(2L+1) \right)^2$$
$$\le 32(2L+1)M^4 + 8(2L+1)\beta^2 \log^2(2L+1) \ .$$

Substituting this bound into the result of Corollary D.6 we get

$$G(\lambda_T) - G(\hat{\lambda}) \le 128M^4 \frac{2L+1}{\beta T^2} + 32\beta \left( \frac{(2L+1)\log^2(2L+1)}{T^2} + 2\log(2L+1) \right) \ .$$

Finally, notice that for all positive integer $L > 0$ it holds that $\log(2L+1) \le \log^2(2L+1)$ and if $T \ge \sqrt{2L+1}$ then we have

$$G(\lambda_T) - G(\hat{\lambda}) \le 128M^4 \frac{2L+1}{\beta T^2} + 32\beta \left( \log^2(2L+1) + 2\log^2(2L+1) \right) \ .$$

$\qquad\square$

Finally, if we set $\beta$ as

$$\beta = M^2 \frac{\sqrt{2L+1}}{T \log(2L+1)} \ ,$$

the bound reads as

$$G(\lambda_T) - G(\hat{\lambda}) \le 256M^2 \frac{\sqrt{2L+1}\log(2L+1)}{T} \ .$$

Thus, in order to achieve an $\varepsilon$ precision, we need to set $T$ as

$$T = \frac{256M^2}{\varepsilon} \sqrt{(2L+1)}\log(2L+1) \ .$$

Our statistical analysis summarized in Theorem 3.3 suggests that $L \sim N^{1/4}$ gives the best convergence rate in terms of the excess risk. Therefore, in order to achieve and $\varepsilon$ precision for the desired minimization it is sufficient to satisfy

$$T \sim \frac{N^{1/8}\log(N)}{\varepsilon} \ .$$

In order to match the rate of convergence for the excess risk an fairness, it is desirable to set $\varepsilon \sim N^{-1/4}$. So the final runtime of our algorithm is $O(N^{3/8} \log(N))$ + the time spent on the construction of $\hat{\eta}$.

# E Algorithm for predictions without sensitive attribute

In this section we propose a modification of our methodology for the case when the predictions are defined as $f : \mathbb{R}^d \to \mathbb{R}$. That is, the fair optimal predictor $f^* : \mathbb{R}^d \to \mathbb{R}$ is defined as a solution of

$$\min_{f : \mathbb{R}^d \to \mathbb{R}} \left\{ \mathbb{E}(Y - f(X))^2 \ : \ \forall \mathcal{C} \subset \mathbb{R} \ \ \mathbb{P}\left(f(X) \in \mathcal{C} \mid S = 1\right) = \mathbb{P}\left(f(X) \in \mathcal{C} \mid S = -1\right) \right\} \ .$$

**Remark E.1.** *In this part of the supplementary material we use the same notation as in the main body. This section should be seen independently from the main body. For instance, the reader should not confuse $f^*$ defined in the main body of the paper and $f^*$ defined above.*

Similarly to the case with the use of $S \in \mathcal{S}$, we work under the bounded signal Assumption 2.3, that is, $|Y| \leq M$. First we define the binned optimal fair predictor $g_L^* : \mathbb{R}^d \to \mathcal{Q}_L$, where $\mathcal{Q}_L$ is the uniform grid on $[-M.M]$ of $2L + 1$ points defined in the main body. The binned optimal fair predictor $g_L^* : \mathbb{R}^d \to \mathcal{Q}_L$ is a solution of

$$\min_{g : \mathbb{R}^d \to \mathcal{Q}_L} \left\{ \mathbb{E}(Y - g(X))^2 \ : \ \forall q \in \mathcal{Q}_L \ \ \mathbb{P}\left(g(X) = q \mid S = 1\right) = \mathbb{P}\left(g(X) = q \mid S = -1\right) \right\} \ .$$

Following the proof of Lemma 2.4 line by line, it is clear that an analogous statement holds in this case. Thus, in order to extend the approach of the main body of this work to the case where the prediction function does not bring into play the sensitive feature, we need to derive the form of $g_L^*$ for all integer $L > 0$.

Let us define[7] $\eta(X) := \mathbb{E}[Y|X]$, $\tau(X) := \mathbb{P}\left(S = 1 \mid X\right)$, and $p_s = \mathbb{P}(S = s)$ for all $s \in \mathcal{S}$.

**Assumption E.2.** *The mappings $t \mapsto \mathbb{P}_X(\eta(X) \geq t)$ and $t \mapsto \mathbb{P}_X(\tau(X) \geq t)$ are continuous.*

**Theorem E.3.** *For each $L > 0$ under Assumption E.2 it holds for all $x \in \mathbb{R}^d$ that*

$$g_L^*(x) = \underset{\ell \in \{-L,\dots,L\}}{\arg\min} \left\{ (\eta(x) - \ell M/L)^2 + \lambda_\ell^* \left( \frac{\tau(x)}{p_1} - 1 \right) \right\} \times \frac{M}{L} \ ,$$

*where $\lambda^* = (\lambda_{-L}^*, \dots, \lambda_L^*)^\top$ is a solution of*

$$\min_\lambda \left\{ \mathbb{E}_X \max_\ell \left\{ \lambda_\ell \left( 1 - \frac{\tau(X)}{p_1} \right) - (\eta(X) - \ell M/L)^2 \right\} \right\} \ .$$

*Proof.* Fix some integer $L > 0$. Notice that we can write for all $g : \mathbb{R}^d \to \mathcal{Q}_L$

$$\mathbb{E}(Y - g(X))^2 = \mathbb{E}_X(\eta(X) - g(X))^2 + \mathbb{E}(Y^2) - \mathbb{E}(\eta^2(X)) \ .$$

Thus, $g_L^*$ can be equivalently defined as a solution of

$$\min_{g : \mathbb{R}^d \to \mathcal{Q}_L} \left\{ \mathbb{E}_X(\eta(X) - g(X))^2 \ : \ \forall q \in \mathcal{Q}_L \ \ \mathbb{P}\left(g(X) = q \mid S = 1\right) = \mathbb{P}\left(g(X) = q \mid S = -1\right) \right\} \ .$$

For an arbitrary $q \in \mathcal{Q}_L$ and $s \in \mathcal{S}$ we can write

$$\mathbb{P}\left(g(X) = q \mid S = s\right) = p_s^{-1}\mathbb{P}(g(X) = q, S = s) = p_s^{-1}\mathbb{E}_X[\mathbf{1}_{\{g(X)=q\}}\mathbb{P}\left(S = s \mid X\right)] \ ,$$

therefore for $(*) = \mathbb{P}\left(g(X) = q \mid S = 1\right) - \mathbb{P}\left(g(X) = q \mid S = -1\right)$ we can write

$$(*) = \sum_{s \in \mathcal{S}} s p_s^{-1}\mathbb{E}_X[\mathbf{1}_{\{g(X)=q\}}\mathbb{P}\left(S = s \mid X\right)]$$

$$= p_1^{-1}\mathbb{E}_X[\mathbf{1}_{\{g(X)=q\}}\mathbb{P}\left(S = 1 \mid X\right)] - p_{-1}^{-1}\mathbb{E}_X[\mathbf{1}_{\{g(X)=q\}}\mathbb{P}\left(S = -1 \mid X\right)]$$

$$= p_1^{-1}\mathbb{E}_X[\mathbf{1}_{\{g(X)=q\}}\tau(X)] - p_{-1}^{-1}\mathbb{E}_X[\mathbf{1}_{\{g(X)=q\}}(1 - \tau(X))]$$

$$= \mathbb{E}_X\left[ \left( \frac{\tau(X)}{p_1 p_{-1}} - \frac{1}{p_{-1}} \right) \mathbf{1}_{\{g(X)=q\}} \right] \ .$$

The above implies that

$$(*) = 0 \Leftrightarrow \mathbb{E}_X\left[ \left( \frac{\tau(X)}{p_1} - 1 \right) \mathbf{1}_{\{g(X)=q\}} \right] = 0 \ .$$

Hence, $g_L^*$ is a solution of

$$\min_{g : \mathbb{R}^d \to \mathcal{Q}_L} \left\{ \mathbb{E}_X(\eta(X) - g(X))^2 \ : \ \forall q \in \mathcal{Q}_L \ \ \mathbb{E}_X\left[ \left( \frac{\tau(X)}{p_1} - 1 \right) \mathbf{1}_{\{g(X)=q\}} \right] = 0 \right\} \ . \tag{20}$$

**Remark E.4.** *Notice that if $X$ is independent from $S$, then $\tau(X) \equiv p_1$ and* any *predictor is fair.*

The rest of the proof is similar to the proof of Proposition 2.6. Let us write the problem in Eq. (20) in its unconstrained form. That is, we would like to solve

$$\min_{g:\mathbb{R}^d \to \mathcal{Q}_L} \max_{\lambda} \left\{ \mathbb{E}_X(\eta(X) - g(X))^2 + \sum_{\ell=-L}^{L} \lambda_\ell \mathbb{E}_X \left[ \left( \frac{\tau(X)}{p_1} - 1 \right) \mathbf{1}_{\{g(X)=\ell M/L\}} \right] \right\} \quad.$$

The objective function of this minmax problem can be equivalently written as

$$\mathbb{E}_X \sum_{\ell=-L}^{L} \left[ (\eta(X) - \ell M/L)^2 + \lambda_\ell \left( \frac{\tau(X)}{p_1} - 1 \right) \right] \mathbf{1}_{\{g(X)=\ell M/L\}} \quad.$$

Now, as before we focus on the dual maxmin formulation of the problem

$$\max_{\lambda} \min_{g:\mathbb{R}^d \to \mathcal{Q}_L} \left\{ \mathbb{E}_X \sum_{\ell=-L}^{L} \left[ (\eta(X) - \ell M/L)^2 + \lambda_\ell \left( \frac{\tau(X)}{p_1} - 1 \right) \right] \mathbf{1}_{\{g(X)=\ell M/L\}} \right\} \quad.$$

The inner minimization problem can be solved explicitly and the solution for all $\lambda \in \mathbb{R}^{2L+1}$ is given by $\tilde{g}_\lambda$ defined for all $x \in \mathbb{R}$ as

$$\tilde{g}_\lambda(x) = \operatorname*{arg\,min}_{\ell \in \{-L,\dots,L\}} \left\{ (\eta(x) - \ell M/L)^2 + \lambda_\ell \left( \frac{\tau(x)}{p_1} - 1 \right) \right\} \times \frac{M}{L} \quad.$$

Substituting the expression for $\tilde{g}_\lambda$ into the objective function of the maxmin formulation we get

$$\max_{\lambda} \left\{ \mathbb{E}_X \min_{\ell} \left\{ (\eta(X) - \ell M/L)^2 + \lambda_\ell \left( \frac{\tau(X)}{p_1} - 1 \right) \right\} \right\} \quad.$$

Let $\lambda^*$ be any minimizer of the above problem. To finish the proof we show that $\tilde{g}_{\lambda^*}$ is fair. It is done similarly to the proof of Proposition 2.6. That is, we first make use of Assumption E.2 to conclude that the objective function in the maximization problem for $\lambda^*$ is almost surely smooth. Then, we write the first order optimality condition for smooth concave maximization problem which precisely gives the fairness of $\tilde{g}_{\lambda^*}$. Thus, $g_L^* = \tilde{g}_{\lambda^*}$ and we conclude. $\square$

**Remark E.5.** *It is straightforward to construct a plug-in method once the form of the optimal predictor is established. Indeed, we only need to solve three problems:*

- *Unconstrained regression on $(X, Y)$, to estimate $\mathbb{E}[Y|X]$.*

- *Unconstrained classification on $(X, S)$ to estimate $\mathbb{P}(S = 1|X)$.*

- *Unconstrained minimization over $\lambda \in \mathbb{R}^{2L+1}$.*

*The statistical analysis of this method is left for future research.*

# F    The Impact of Unlabeled Data on the Performance of the Estimator

In this section, we empirically study the behavior of the proposed estimator as a function of unlabeled data sample used for recalibration. For this purpose, since the benchmark datasets considered in this paper are fully labeled, we subsample from the original dataset a smaller labeled sample $\mathcal{D}_n$ and then simulate a scenario in which the unlabeled sample $\mathcal{D}'_N$ varies. Specifically, we choose $n = 1/10$ the size of dataset used to estimate $\eta$, and $N \in \{0, 1/10, 2/10, 4/10, 8/10\}$ the size of the dataset considered to recalibrate $\eta$ as a fair predictor. This data generation procedure is applied to the LAW dataset, since it is the largest dataset. We apply our method using the random forest algorithm, using the same cross-validation scheme as in Section 4. The above pipeline is repeated 30 times and the variance of the results is reported in Table 2. Notice that both MSE and DDP are improving with $N$, highlighting the importance of the unlabeled data. We believe that the improvement could have been more significant if the unlabeled data were provided initially.

| LAW - RF+Ours | MSE | DDP |
|---|---|---|
| $\mathcal{D}_n = {}^1\!/_{10}$ | $.096 \pm .012$ | $.046 \pm .005$ |
| $\mathcal{D}_n = {}^1\!/_{10}, \mathcal{D}'_N = {}^1\!/_{10}$ | $.093 \pm .011$ | $.044 \pm .005$ |
| $\mathcal{D}_n = {}^1\!/_{10}, \mathcal{D}'_N = {}^2\!/_{10}$ | $.092 \pm .010$ | $.041 \pm .005$ |
| $\mathcal{D}_n = {}^1\!/_{10}, \mathcal{D}'_N = {}^4\!/_{10}$ | $.090 \pm .010$ | $.039 \pm .005$ |
| $\mathcal{D}_n = {}^1\!/_{10}, \mathcal{D}'_N = {}^8\!/_{10}$ | $.089 \pm .010$ | $.038 \pm .004$ |

Table 2: Impact of the size of the unlabeled dataset on MSE and DDP. The size of the labeled sample $\mathcal{D}_n$ is fixed to $1/10$ of the original dataset size. The unlabeled $\mathcal{D}_N$ is initially empty (meaning that we both estimate $\eta$ and recalibrate it using the same sample $\mathcal{D}_n$, as in the previous experiments of Table 1), and then it increases from $1/10$ to $8/10$ of the original dataset.