[Reviews · NeurIPS 2020]

Review 1

Summary and Contributions: Summary: This paper introduces an approach to incorporate fairness constraints, specifically demographic parity, into least squares regression tasks. It’s based on a two-step scheme: first run any standard risk minimisation regression algorithm without fairness constraints, then enforce approximate demographic parity based on a discretisation of the target predictions and a suitable adjustment of the regression function on the resulting discretised grid. This adjustment requires the solution of a discrete but convex optimisation problem which is solved with a smoothed accelerated gradient method. Finite sample convergence bounds are presented both for approximations of the fairness and risk criteria. Experimental results on a variety of datasets fare comparably or favourably against a number of methods, including recently proposed techniques that arguably well represent the state-of-the-art. Contributions: A new method for fair regression which consists of post-processing a standard unconstrained regression in a way so as to promote demographic parity Finite sample convergence bounds (for both fairness and risk estimates) providing insight into rates of convergence and how to optimally set hyperparameters Experimental study on various datasets comparing the proposed with a number of state-of-the-art approaches

Strengths: This paper has a number of strengths: It addresses a relevant problem for the NeurIPS community, which has been given relatively little attention in the algorithmic fairness literature: how to quantify and promote algorithmic fairness in regression settings (the vast majority of the work is restricted to classification settings) It is clearly written, well organised and easy to read in spite of its significant technical density It provides a conceptually simple, sound and modular approach that allows the use of any regressor for risk minimisation and only promotes demographic parity as a post-processing adjustment The algorithmic solution for the reduction proposed is clean and well justified theoretically Theoretical uniform convergence rates are provided for both fairness and risk estimates Experiments cover a good range of datasets and compare similarly or favourably against a couple of recent approaches dedicated to fair regression (Oneto et al and Agarwal et al in particular)

Weaknesses: I’m hesitant to flag “weaknesses” as I feel this is a very solid and well-rounded paper. I have provided some feedback and added specific suggestions for improvement in the corresponding section below.

Correctness: The technical claims and the experiments are credible. The evidence provided is in line the claims made. There is nothing in the empirical methodology that seems flawed.

Clarity: I found the paper well-structured, clear and easy to read (I’d say above average for highly technical papers of this kind).

Relation to Prior Work: Yes, the relation to technical work seems appropriate to me. E.g. the recent approaches from Agarwal et al and and Oneto et al arguably represent good samples from the state of the art, and they were referred to and compared against in the experiments ( as well as earlier relevant approaches such as Berk et al).

Reproducibility: No

Additional Feedback: I have read the author response other reviews. I think this is a strong paper, providing both a practical algorithm and theoretical guarantees. Some discussion of whether or not the approach could be extended to an equal opportunity metric - the most obvious candidate is P( f | Y, S) - would be welcome but I think the paper is a sufficient contribution as it stands. The experimental results were shown only for the optimal selection of L according to the risk bound. However, in practice it’s important to have flexibility on how to balance accuracy and fairness objectives, as the optimal trade-off can vary across different application contexts. It would be beneficial to generate accuracy-fairness Pareto frontiers and compare that against the other methods, to better understand from an empirical perspective the regimes in which the proposed approach is particularly stronger/weaker


Review 2

Summary and Contributions: This paper provides a new algorithm to train a regression function subject to a demographic parity like fairness constraint. The proposed approach constructs a plug-in estimator by first training an unconstrained regression function using labeled data and calibrate the model to satisfy the fairness constraint using unlabeled data. The final model is a "regression function with discrete outputs". The authors show convergence rates to the optimal fair regression model, and demonstrate competitive empirical performance compared to previous approaches for fair regression. ################################################################## POST-REBUTTAL: Thanks a lot for the detailed response. I'm still of the opinion that the technical gap I pointed out is an important one, and that the analysis would have been much more complete and satisfying had the guarantees for the optimization algorithm been on the gradients of the dual objective. **I strongly urge the authors to at least include a discussion on this gap in the main paper**. I think one of the reasons it may actually turn out to be difficult to provide convergence guarantees on the dual gradient (without additional strong assumptions) is because the dual objective G(.) is non-smooth. As you yourself point out, your current proof is only able to control the gradient of the smoothed dual objective. This is one the reasons why I think prior approaches (e.g. Agarwal et al.) provide statistical guarantees assuming an *approximate saddle point*, instead of near-zero gradient. I'm also sorry that I missed out a couple of references in my initial review. Please see: https://arxiv.org/pdf/2006.13485.pdf (Table 2), and the references there in for prior attempts at the use of plug-in estimators for imposing fairness constraints. Here I think the authors consider a multiclass problem (but might be closely-related as you essentially reduce the regression problem to a multiclass problem via discretization). However, these prior works do not particularly exploit the structure of the plug-in classifier as you do, something I liked about the present paper. ##################################################################

Strengths: 1) A key advantage is the use of a plug-in approach, that allows one to employ any off-the-shelf regression algorithm, and simply post-process the trained model with a relatively less expensive procedure. Only the initial regression learning step requires labeled data, whereas the second step can use unlabeled data. 2) I like how the authors exploit the structure of the plug-in regressor to apply accelerated gradient descent to optimize over Lagrange multipliers. Prior works that have similarly used a plug-in approach for fair classification problems (e.g. ref 1-2 below) do not exploit this additional structure and are thus able to achieve a better O(1/\epsilon) convergence rate to the optimal multipliers. 3) Unlike prior work by Agarwal et al., the proposed method does not learn a "stochastic" model (i.e. a distribution over multiple regression functions), making the approach more practical. Usually stochastic models are used for constrained training problems in order to "convexify" the search space, but interestingly, the authors have shown that for the specific case of plug-in estimators with demographic parity constraints, deterministic classifiers suffice! (I do however have a question later on whether the convergence guarantees are robust to errors in parameter estimates).

Weaknesses: 1) Discrete regression outputs: The output space for the final regression model is *discrete*, and this may not be entirely desirable in applications where the regression function is expected to make continuous predictions. I understand that prior works on fair regression also discretize the regression problem to turn it into a classification problem, but this could be a limitation in practice. 2) Feasibility result doesn't take errors in multipliers into account: There is a gap in the theoretical results presented. The main optimality and feasibility results in Sec 3.1 assume that the Lagrange multipliers \lambda's are exactly *optimal*. However, the multipliers output by Algo 1 achieve a dual objective that is "\epsilon-close" to the optimal dual objective. Are your feasibility and optimality results robust to errors in the multipliers? In particular, your proofs suggest that you would need the multipliers to have a near-zero dual gradient, whereas you guarantee that the multipliers output by Algo 1 are close-to-optimal in objective value. 3) Can you handle the more common case of inequality constraints with slack? The current definition of a "fair predictor" (Defn 2.1) insists that the output distribution for each sensitive attribute be equal, and as a result the authors end up with "equality" fairness constraints in their formulation. However, in practice, its more reasonable to allow for a slack in the fairness constraints (so that one can better control the trade-off with the MSE objective) and turn them into inequality constraints. How important is it that the fairness constraints in your formulation be equality constraints? With inequality constraints, you would additionally need the Lagrange multipliers to be non-negative, and as a result may not be able to simply equate the gradient of the dual objective to 0 to show feasibility. Moreover, the accelerated GD algorithm would need to explicitly impose non-negativity constraints on the multipliers (e.g. using a prox term). Overall, the paper's strengths outweigh the flaws. I'll be happy to adjust my scores based on the authors' response to question (2) in particular. Finally, while the authors already consider a number of baselines, for whatever its worth, here's one more post-processing baseline that I've found to work very well in practice for regression problems. [3] Jiang et al. "Wasserstein Fair Classification", UAI 2019. http://auai.org/uai2019/proceedings/papers/315.pdf While the problem setup here is that of class probability estimation, with a [0,1] continuous output space, the post-processing algorithm in Algo 2 in their paper is usually very effective for imposing demographic parity constraints even for general real-valued regression outputs. Instead of equalizing the output distribution between two groups, this paper aims to match the output distribution for the each group with the fixed distribution of labels P(Y = \ell) in the data. While this is a more restrictive constraint, it allows for a simpler quantile-based post-processing step. Minor comments: - In the experimental comparisons, how do you set the slack \epsilon for the constraints when you ran the baseline methods (e.g. Agarwal et al.)? Unless you fix the same slack on the KS violation for all the methods, it becomes hard to perform a fair comparison. - You currently report the constraint violation (KS metric) on the test set. How does this metric look like for your method on the train set -- in the absence of generalization error, to what extent does the discretization done in Sec 2.2. effect the violation in constraints? - In the proof of Prop 2.6, is it obvious why strong duality holds for the min-max problem despite the indicator function line 522? In particular, when you say it is easy to see that R(˜gλ∗ ) = R(g*) (line 535), wouldn't this require interchanging the min and max (or at least a proof similar to Lemma C.2.) - In the discussion of VC theory in Appendix B and the results that follow, my understanding is that you turn the problem of predicting an output in a range \mathcal{C} \subset \mathbb{R} into a binary classification problem, and bound the VC-dimension of the regressor-turned-classifier. Please correct me if my understanding is wrong. Might help the reader if you provide some intuition early on in this section in plain text.

Correctness: Yes, to the extent I could verify the proofs in the appendix.

Clarity: Yes, the paper is nicely written.

Relation to Prior Work: Yes, to the extent I could verify.

Reproducibility: Yes

Additional Feedback: Encourage the authors to release the code for their algorithm.


Review 3

Summary and Contributions: The paper addresses the problem of fairness in regression. The goal of the paper is to train regression models that satisfy the equivalent of demographic parity notion of fairness in regression. The paper uses a fairly strong measure of fairness, which requires equality of representation in all the bins of the regression output. Training of the fair regression model is done by taking a pre-trained (unfair) regression model and discretizing the output space. Next, the unlabeled data is used to estimate different marginal distributions and the pretrained regression model is "calibrated" to produce fair scores.

Strengths: 1- The biggest strength of the paper is the principled approach towards solving the problem. Each assumption (e.g., discretizing the regression score) is explained well and the effect of choices on the statistical guarantees is shown. 2- The paper is well-written, and does a good job of walking the reader through the design choices. 3- The empirical results also support the proposed method quite well. 4- It is quite nice that the fairness bound in Theorem 3.2 does not depend on the quality of the learnt estimator. This is quite different from much of the prior work on post-processing which take this estimator to be the Bayes optimal estimator. As Bayes optimal estimators are quite hard to obtain in practice, the fairness and accuracy guarantees of the estimators can be somewhat confusing.

Weaknesses: While the paper has unquestionable strengths, there is also an area where an improvement, or at the very least, some comments are needed: In last 4-5 yeas, the fairness literature has gravitated more and more towards the equality of opportunity notion of fairness. This shift happened as demographic parity might be very strong notion of fairness in many situations (e.g., requiring that equal fraction of applicants from different demographic groups be given the desirable outcome, regardless of the qualifications). Equality of opportunity is able to overcome some of these issues. So it is a bit unfortunate that the paper does not cater to equality of opportunity. This in itself is perhaps not a deal-breaker, but at least some comment here would definitely be helpful. Specifically, how would the paper go about defining equality of opportunity for a regression scenario, and how could the machinery of the paper be adapted for this notion.

Correctness: I only skimmed over the proofs in the supplemental section, so cannot vouch for the correctness there. The experimental section however seems well-thought through, and compares against a number of baselines, so no complaints there.

Clarity: Yes, the paper is in fact quite well written. Terminology is laid out very clearly and the design choices are explained very well.

Relation to Prior Work: Yes, the connections/differentiations to the prior work are quite clearly made.

Reproducibility: Yes

Additional Feedback: 1- How is the runtime of the proposed method as compared to more simpler schemes such as Berk et al. A high runtime is not a big problem, however, it would be good for the reader to know this. 2- How big / small is C in Theorem 3.2? 3- The proprietary data that the paper collects seems to contain ethnicity feature as well. Was this feature also a part of the learning task? It is clear that gender was taken as the sensitive feature, but what about ethnicity?


Review 4

Summary and Contributions: This paper proposes an algorithm to achieve fairness of regression function. The proposed method can utilize labeled data and unlabeled data. Also, the error rates and fairness of the proposed estimator are proven to be converged. The experiments in this paper show that the proposed algorithm can achieve fairness while still competitive with state-of-the-art methods. I have read authors' rebuttal, and maintain my accept rating.

Strengths: The theoretical foundation of this paper is strong: the fairness and error rate are guaranteed to be converged. The proofs are provided in supplementary material exhaustively. I checked some of these proofs and found no error.

Weaknesses: A minor weakness is that the proposed algorithm sacrifices a small amount of error rate to achieve fairness.

Correctness: The proofs in supplementary materials and experiments are robust to demonstrate these claims are correct.

Clarity: This paper is well-written. Also, every math symbol is introduced in clarity.

Relation to Prior Work: Sufficient prior works are introduced to compare with this work. Their methods are compared with these works in the experiments.

Reproducibility: No

Additional Feedback: In Table 1, the lowest KS and lowest MSE in each dataset should be highlighted for clarity. The link of source code is provided in the footnote of page 7; however, this link is anonymous. The source code should also be provided in the supplementary material.

[Author Response · NeurIPS 2020]

We thank all reviewers for their positive and valuable comments. Let us first address comments shared by multiple reviewers and then answer individual comments.

**Relaxation of fairness constraints.** In our contribution the smoothing parameter $\beta$ is introduced to accelerate subgradient methods, however during our experiments we observed that it also controls risk/fairness trade-off. Hence, a relaxation of fairness constraints is implicit in our method via $\beta$. It would be interesting to derive statistical theory connected to this parameter in future works. We will add an illustration highlighting this phenomena. Another direction that one can take is to directly relax the fairness constraints asking for $\max_{q \in \mathcal{Q}_L} \mathcal{U}(g, \{q\}) \leq \varepsilon$ (see ll. 114–115). Our methodology is flexible enough to allows one to deal with this case and our theory can be adjusted accordingly as well. The price for such relaxation is doubling the dimension of the Lagrange multipliers due to the absolute value in the definition of $\mathcal{U}$. We will sketch the argument in the appendix.

**Other notions of fairness.** It would indeed be interesting to study extensions of our techniques to other notions of fairness. Though, to the best of our knowledge, there is no general notion of Equalized Odds in regression that would be widely accepted in literature, the main technical difficulty with a possible extension of our machinery to such notion would come from the conditioning on $Y$.

**R1 Visual description.** To help the reader we will add the plot to the right, illustrating the distribution of the prediction for $L{=}25$ on CRIME.

Values assumed by f

**R2 Discrete regression outputs.** We agree that in some practical applications the discrete outputs might be undesirable. First of all note that for sufficiently large discretization parameter $L$, the discrete nature of the output might be unnoticeable. Secondly, some classical fairness unaware methods such as kNN and decision trees also construct only discrete outputs. Yet, if discrete prediction is still an issue for the problem at hand, a simple ad-hoc remedy is to first fit our method, and then construct an *interpolating* curve (for instance polynomial of sufficiently high degree) using an additional unlabeled dataset. **Errors in multipliers into account.** Let us point out that it is a standard practice in statistical theory to separate the statistical analysis from the numerical optimization problem. Nevertheless, as reviewer pointed out, in order to carry out our proof the dual gradient should be controlled. To this end, note that $G_\beta$ defined at line 754 is convex and has $2/\beta$-Lipschitz gradient. Thus $\|\nabla G_\beta(\lambda_T)\|_2^2 \leq (\beta/4)\{G_\beta(\lambda_T) - \min G_\beta(\lambda)\}$. Furthermore, Thm. D5 in the appendix provides a control on $\{G_\beta(\lambda_T) - \min G_\beta(\lambda)\}$, hence we can control $\|\nabla G_\beta(\lambda_T)\|_2^2$ – the gradient of the smoothed objective. Finally, note that the gradient of $G(\cdot)$ is essentially the argmax function, while the gradient of $G_\beta(\cdot)$ is the soft-arg-max (see Lem. D4). Controlling the deviation of the gradient of $G(\cdot)$ from the gradient of $G_\beta(\cdot)$ would yield the desired bound. Even though this extension is interesting, we feel that it would further complicate already dense proofs with little practical benefits. **Wasserstein fair classification.** Though this work deals with classification and does not tackle the problem of risk minimization under Demographic Parity (DP), we thank the reviewer for this relevant reference. **How do you set slack variable?** All the hyperparameters are tuned according to the scheme described in ll. 249–255. **How does discretization effect violation of constraint?** Actually, the less atoms in discretization, the easier it is to be fair in the sense of DP. Imagine for simplicity that the discretization consists of one point, then the discretized predictors set $\mathcal{G}_L$ (see ll. 113–114) contains only one constant function, which is of course fair. However, if there are too few atoms in the discretization, then the corresponding predictors in $\mathcal{G}_L$ are not powerful enough to yield a good risk guarantee (Lem. 2.4). Thus, the discretization should be balanced between the risk and fairness. **VC-theory.** The intuition of the reviewer is correct. We essentially transform the problem into a *multi-class classification* setup with a specific risk. The main difficulty of course is to understand the amount of classes that one should pick. **Exchange of min and max.** In principle the reviewer is correct that the claim that $\mathcal{R}(\tilde{g}_{\lambda^*}) = \mathcal{R}(g^*)$ is equivalent to strong duality. The idea of the proof is to actually establish that this strong duality holds. Note that since $\tilde{g}_{\lambda^*}$ solves the *dual* problem, then $\min\{\mathcal{R}(g) : g \text{ is fair}\} \geq \mathcal{R}(\tilde{g}_{\lambda^*})$ thanks to the weak duality. To derive the equality note that at line 534 we demonstrate that $\tilde{g}_{\lambda^*}$ is fair. It means that $\tilde{g}_{\lambda^*}$ is feasible for the primal problem, and then $\min\{\mathcal{R}(g) : g \text{ is fair}\} \leq \mathcal{R}(\tilde{g}_{\lambda^*})$. Thus, $\tilde{g}_{\lambda^*}$ minimizes the primal problem as well. The proof is concluded.

**R4 runtime.** We note that since our algorithm works in post-processing manner, the most demanding part is the training of the base estimator, which largely depends on the considered algorithm. The runtime of the post-processing Algorithm 1 is present at lines 230-234, which is actually sublinear in the amount of data. In contrast, the algorithm of Agarwal et. al (see their Thm. 1 and Alg. 1) is super linear in the input data and each iteration of their algorithm requires to solve two optimization problems. **Constant C.** This constant is exactly the one appearing in Thm. B6. Unfortunately, we did not find a reference with explicit constant. Our rough computations show that it is at most 36. **The proprietary data [...] what about ethnicity?** The data is predominantly mononational, only less then 1% of students are not of the dominant nation. Due to privacy reasons we cannot disclose the dominant nation of the students.

**R5 sacrifices a small amount of error.** Note that since we do not minimize the error over *all* predictions, but only over fair ones, the decrease in accuracy is inevitable unless the regression function is fair in the first place. That is, in our case the sacrifice is due to a very effective satisfaction of the fairness constraints.

[Meta-Review · NeurIPS 2020]

All reviewers agree that this is a solid piece of work, with a theoretically well-motivated proposal for getting a fair regressor from an unconstrained estimator. Please follow the advice of Rev #2 by accompanying the theoretical results with comments explaining their precise context of validity (e.g. exact optimality of Lagrange multiplier), to avoid future misuse of your results.